# Cloud Chamber Studies on the Linear Depolarisation Ratio of Small Cirrus Ice Crystals

Adrian Hamel[1], Martin Schnaiter[1,*], Masanori Saito[2], Robert Wagner[1], and Emma Järvinen[1,*]

[1]Institute of Meteorology and Climate Research Atmospheric Aerosol Research (IMKAAF), Karlsruhe Institute of Technology, Karlsruhe, Germany
[2]Department of Atmospheric Science, University of Wyoming, Laramie, USA
[*]Now at Institute for Atmospheric and Environmental Research, University of Wuppertal, Wuppertal, Germany

**Correspondence:** Adrian Hamel (adrian.hamel@kit.edu) and Emma Järvinen (jaervinen@uni-wuppertal.de)

**Abstract.** Space-borne lidar, in combination with other remote sensing instrumentation, has been used to infer vertical profiles of ice cloud properties from A-train satellites, and more recently, also from the newly launched EarthCARE mission. However, accurately retrieving ice crystal microphysical properties from lidar signals requires a thorough understanding of their relationship to backscattering characteristics. Cloud chambers can be used to study the link under a controlled environment.

This study investigates the link between the linear depolarisation ratio in the near-backscattering direction (178°) and the ice microphysical properties for 47 cloud experiments at cirrus temperatures between -75 °C and -39 °C. Predominantly small (diameter $< 70\,\mu m$) columnar and irregularly shaped ice crystals were grown under distinct conditions of supersaturation with respect to ice. A statistical and visual analysis of size, shape and morphological complexity reveals that more than 40 % of the columnar particles exhibit hollowness on the basal facets. Ice crystals larger than $10\,\mu m$ show depolarisation ratios below

0.3, which is lower than typical values observed in mid-latitude cirrus but in agreement with polar cirrus observations. Two temperature-dependent depolarisation ratio - size modes were found and successfully reproduced with ray tracing simulations of hollow columns incorporating surface roughness, hollowness and internal scattering. These results are important for the interpretation of the linear depolarisation ratio of small ice crystals in active remote sensing or can be used for evaluating the performance of state-of-the-art optical particle models, especially for small size parameters below 100.

## 1 Introduction

Depolarisation lidar measurements are useful to identify the presence of ice clouds because the non-spherical ice particles alter the polarisation during the scattering process (Liou and Lahore, 1974). Furthermore, size and shape affect the linear depolarisation ratio of ice crystals, but the link is highly complex (Sassen, 1991). Therefore, retrieving cirrus microphysical properties using the linear depolarisation ratio is challenging. Sassen and Zhu (2009) showed a decrease in the linear depolarisation ratio

for increasing altitude and decreasing temperature using two-year global linear depolarisation ratio data of ice clouds measured by the Cloud-Aerosol Lidar with Orthogonal Polarization (CALIOP) onboard the Cloud-Aerosol Lidar and Infrared Pathfinder Satellite Observations (CALIPSO) satellite. Sassen et al. (2012) found a strong correlation between temperature and linear

depolarisation ratio. This highlights that models using a vertically homogenous ice crystal shape model are inappropriate for radiative transfer calculations.

Numerical studies have suggested that it is possible to use linear depolarisation properties to infer ice crystal shape information. For instance, Noel et al. (2002) introduced a shape classification technique for hexagonal ice crystals where higher depolarisation ratios are associated with columns (higher aspect ratios) and low depolarisation ratios with plates (lower aspect ratios). Assuming pristine ice crystals, they identified four classes of aspect ratios from the linear depolarisation ratio based on ray tracing simulations. In addition, global cirrus cloud particle habit fractions were derived from CALIOP satellite LIDAR

data using the physical particle model, an improved geometric optics ray tracing methods which includes multiple scattering effects (Sato and Okamoto, 2023). Not only shape but also size information is suggested to be inferred from lidar backscattering depolarisation measurements (Kustova et al., 2022). The ray tracing simulations of pristine columns over a size range from 10 to 1000 μm in the geometric optics approximation showed that the linear depolarisation ratio for hexagonal ice crystals oscillates with the ice crystal size. For absorbing wavelengths where the ice crystals are not transparent (e.g. 2 μm), the

absorption also decreases the linear depolarisation ratio with size.

    However, the applicability of these numerical results produced with idealized crystal geometry to atmospheric remote sensing observations is highly uncertain because recent studies have observed that cirrus ice crystals rarely show idealized hexagonal shape and almost always contain some degree of morphological complexity (Järvinen et al., 2023). Saito and Yang (2023) used the Invariant Imbedding T-Matrix (IITM) and the improved geometric optics method (IGOM) to model the effects of

both size and surface roughness on the linear depolarisation ratio of hexagonal ice crystals with an aspect ratio of one. They concluded that the ice crystal roughness has a strong impact on the backscattering properties, which needs to be included in order to simulate observational data. Adding hollowness to the basal facets of bullet rosette ice crystals was found to lower the backscattering linear depolarisation ratio using improved geometric ray tracing simulations (Yang et al., 2008). Furthermore, a decreasing trend in linear depolarisation ratio for increasing particle size was seen that was not present for solid bullet rosettes.

Cloud chamber experiments have proved useful for studying the link between ice morphological and optical properties under well-defined atmospheric conditions (e.g. temperature or ice saturation ratio). Previous studies have shown high linear depolarisation ratios in the near-backscattering direction (178°) of up to 0.4 for sublimating small ice crystals (<10 μm) at a temperature range between -50 °C and -70 °C (Schnaiter et al., 2012). A cloud chamber study conducted at a higher temperature range between -7 °C and -30 °C by Smith et al. (2016) showed large discrepancies between the measured linear depolarisation

ratio and simulations assuming idealized pristine hexagonal shapes for larger ice crystals (20 μm < maximum dimension < 200 μm) for the near-backscattering (178°) and the backscattering (180°) directions. These discrepancies could be reduced by assuming stepped hollowness and by using a tilted facet method in the ray tracing simulations to account for ice crystal complexity.

    This study advances previous work on the relationship between ice microphysical properties and linear depolarisation ratio

by providing a more statistically robust analysis, based on a multi-year dataset from four laboratory cloud chamber campaigns. In total, 47 ice cloud simulation experiments were conducted under well-controlled conditions at cirrus temperatures between –39°C and –75°C. A key strength of this study is the ability to control ice crystal growth under specific, known temperature

and supersaturation conditions, enabling a direct assessment of how particle size, shape, and optical complexity influence depolarisation signals. In addition, the measurements are compared with results from conventional as well as state-of-the-art light scattering models. Together, this approach provides new insight into the microphysical drivers of depolarisation and offers improved constraints for the interpretation of active remote sensing observations of cirrus clouds.

The microphysical and optical instrumentation used to measure the microphysical and optical properties of the cloud chamber grown ice particles are introduced in sections 2.1 and 2.2. The experimental procedures and the numerical simulations are detailed in sections 2.3 and 2.4. Section 3.1 contains a detailed analysis of the microphysical properties, such as size, small-scale complexity and hollowness, of the cloud chamber grown ice particles. Hereafter, the microphysical properties size (section 3.2) and small-scale complexity (section 3.3) are linked to the linear depolarisation ratio, with the main outcome being the evolution of the linear depolarisation ratio as a function of the particle size. Section 3.4 compares the experimental results to different numerical T-matrix and ray tracing light scattering simulations. In section 4, the cloud chamber measurements are compared to previous cloud chamber and atmospheric observations and the limitations of the different numerical simulations and of the ice particle morphology representation are discussed. A summary of the study is provided in section 5.

## 2   Methods

In this section, details about the experimental setup and procedures are provided, including the cloud chamber and the optical and microphysical instrumentation that was operated during the experiments. Furthermore, the numerical simulations that are used as a comparison to the experimental results are described.

### 2.1   Optical instrumentation

The linear depolarisation ratio ($\delta$) of an ensemble of ice particles in the cloud chamber is measured with the SIMONE (Streulichtintensitätsmessungen zum optischen Nachweis von Eispartikeln - Scattering Intensity Measurements for the Optical Detection of Ice Particles) instrument (Schnaiter et al., 2012). A schematic setup is shown in Fig. 1a. The instrument measures the scattered intensity in the near forward direction (2°) and in the near-backscattering direction (178°) from a 488 nm continuous wave laser beam propagating through the cloud chamber. Well-mixed conditions and random particle orientations persist due to operating a mixing fan in the cloud chamber. The detection volume of approximately $7\,\mathrm{cm}^3$ is defined by the overlap between the laser beam and field of view of the detector. The laser beam is linearly polarised and the polarisation axis can be rotated with a liquid crystal rotator. For this study, a polarisation parallel to the scattering plane is chosen. In the 178° direction the scattered light is split with a polarising beam splitter and the resulting co- and cross-polarised states are analysed using two photomultiplier tubes (Perkin Elmer MP-1383). The response of the two photomultiplier tubes are calibrated for each campaign using a Spectralon target with a known linear depolarisation ratio that can be placed in the scattering center inside the cloud chamber (Schnaiter et al., 2012). For the RICE03 campaign, data from a different instrument with the same operation principle (SIMONE-Junior), an emission wavelength of 552 nm and a detection volume of approximately $30\,\mathrm{cm}^3$ were used, which is described in Järvinen (2016).

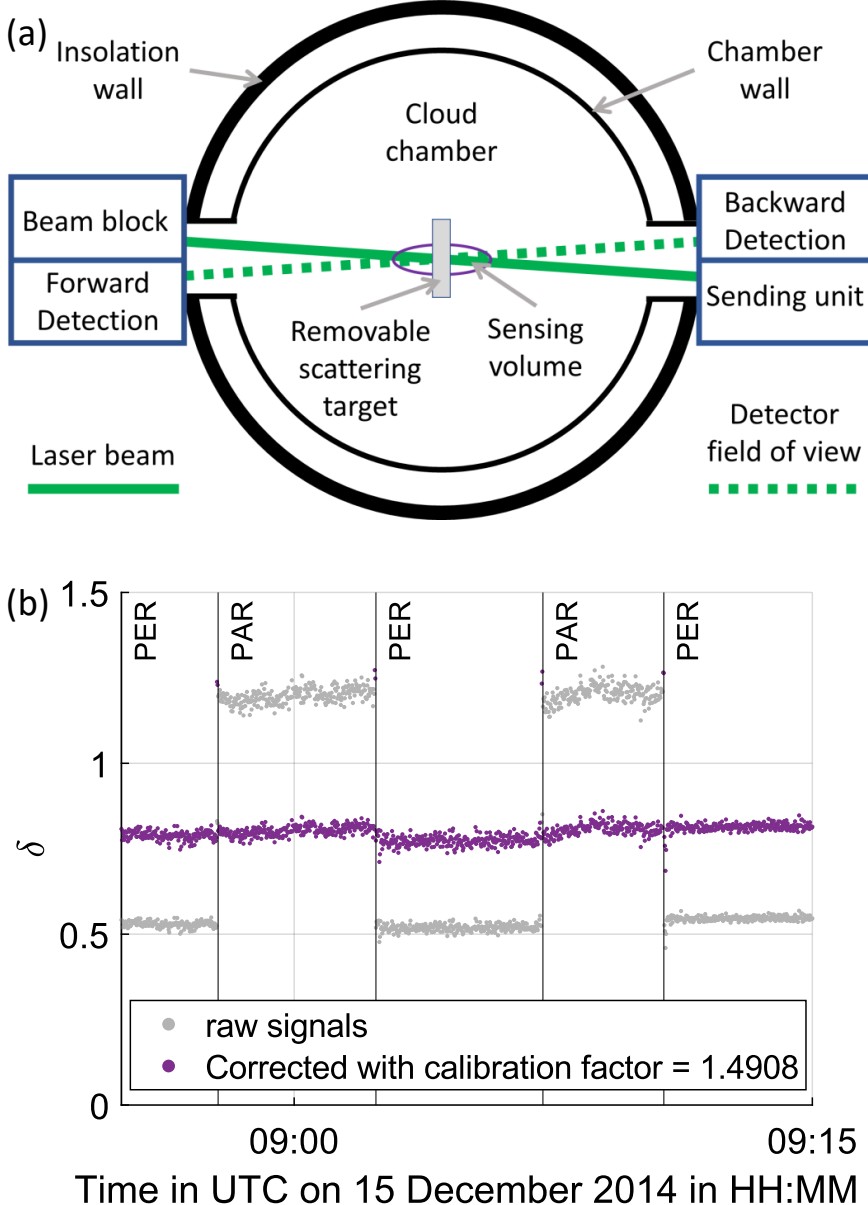

**Figure 1.** Schematic setup of the SIMONE instrument in top view **(a)** and calibration process of SIMONE-Junior for the RICE03 campaign with a scattering target inside the sensing volume **(b)**. PAR refers to incident polarisation parallel to the scattering plane and PER refers to linear polarisation perpendicular to the scattering plane. The calibration factor is determined using the linear depolarisation ratio of the scattering target which is independent of the polarisation direction of the incident light.

During the calibration process, which is identical for SIMONE and SIMONE-Junior, the polarisation of the incident light is switched between parallel and perpendicular orientation with reference to the scattering plane. A calibration factor is multiplied

to the channel, which records the intensity with parallel polarisation, in order to obtain the same linear depolarisation ratio for the linearly polarised incident light at both orientations. Fig. 1b shows the calibration process exemplary for measurement campaign RICE03, where a calibration factor of 1.4908 was derived for SIMONE-Junior. The calibration factor takes into account the different gains of the detectors, different losses of the polarisation filtering and effects of possible differences in alignment of the detectors. The measurement uncertainty of $\delta$ is 2.1 %, derived as the standard deviation of the linear depolarisation ratio of the scattering target measured during the calibration process. Experiments with supercooled liquid droplets were conducted at the AIDA cloud chamber to validate the measurement uncertainty because the linear depolarisation ratio of liquid and thus spherical droplets vanishes (Liou and Lahore, 1974). The mean linear depolarisation ratio of 12 experiments of supercooled liquid clouds at an initial gas temperature of -30 °C is (0.7±0.3) %. The derivation from the theoretical value of 0 % is well below the measurement uncertainty of 2.1 % derived from the calibration with the scattering target. Additional information about the supercooled liquid droplet experiments is provided in appendix C.

The linear depolarisation ratio for incoming light with a polarisation parallel to the scattering plane $\delta$ is calculated as (Mishchenko and Hovenier, 1995):

$$\delta = \frac{I_\perp - I_{\perp,\mathrm{bg}}}{I_\| - I_{\|,\mathrm{bg}}} \tag{1}$$

where $I_\perp - I_{\perp,\mathrm{bg}}$ is the background-subtracted light intensity with a polarisation perpendicular to the scattering plane and $I_\| - I_{\|,\mathrm{bg}}$ with a polarisation parallel to the scattering plane. The linear depolarisation ratio from the SIMONE instrument is averaged over time intervals of 10 s. Hereafter we refer to it as $\delta$. Based on an estimation of the optical depth from the particle microphysics measurements, below 10 % of the intensity detected with SIMONE can be affected by multiple scattering effects for 89.9 % of the measurement data.

## 2.2 Microphysical instrumentation

The particle size, shape and small-scale morphological complexity are optically characterised using the Particle Phase Discriminator 2 Karlsruhe edition (PPD-2K) and the Small Ice Detector 3 (SID-3) (Kaye et al., 2008; Ulanowski et al., 2014; Vochezer et al., 2016; Schnaiter et al., 2016). Both instruments are optical particle counters that detect the scattered light of individual particles. A sample air flow passes the measurement volume that is defined by the overlap between the laser beam with a wavelength of 532 nm from a frequency-doubled Nd:YAG laser and the field of view of the trigger optics. The design of the trigger optics differs in both instruments. In PPD-2K, a beam splitter diverts 8 % of the forward scattering light to the trigger detector. SID-3 uses two nested trigger detectors with half angles of 9.25° at an angle of 50° to the forward scattering direction. In both instruments, the trigger intensity is detected with a photomultiplier tube and used to determine the size of the individual particles with a maximum count rate of 11 kHz. A special feature of SID-3 and PPD-2K is that they use an intensified Photek ICCD218 camera to record the spatial intensity distribution of the forward scattered light over an annulus between approximately 5° and 26° with a resolution of 780×592 pixels (SID-3) and between 7.4° and 25.6° with a resolution of 582×592 pixels (PPD-2K). Images are taken for a subsection of the triggered particle events due to the camera's maximum imaging rate of 30 Hz. These diffraction patterns contain information on the particle habit and morphological complexity at

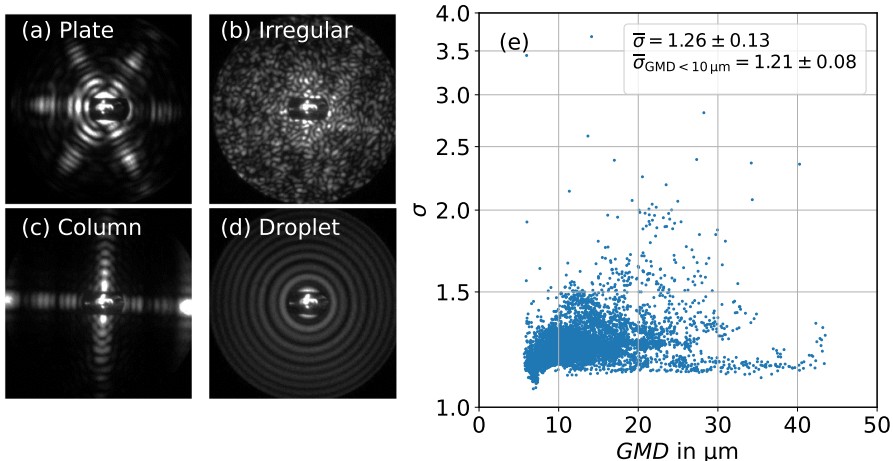

**Figure 2.** Example diffraction patterns of PPD-2K from the RICE01 campaign are shown for a plate **(a)**, an irregular particle **(b)** a column **(c)** and a sphere (droplet) **(d)**. The standard deviation $\sigma$ of the logarithm of the particle diameter is shown as a function of geometric mean diameter ($GMD$) for log-normal fits to the particle size distributions at temperatures below -39 °C **(e)**.

scales of the wavelength of the used light. Both instruments were operated simultaneously. Here we use PPD-2K for getting the size information due to higher counting statistics compared to SID-3 (Vochezer et al., 2016) and SID-3 for getting information on the crystal shape and degree of morphological complexity, similar to Schnaiter et al. (2016).

PPD-2K measures the trigger intensity $I$ of each particle. To relate $I$ to the particle size, the spherical equivalent diameter $d$ is used, which is the diameter of a sphere that scatters the same intensity in the direction of the PPD-2K trigger field of view

at polar angles between 7.4° and 25.6° as the recorded particle. $d$ is calculated from the trigger intensity $I$ using the following equation (Cotton et al., 2010):

$$d = a \cdot I^b \tag{2}$$

where calibration coefficient $a$ depends on the laser power and PMT gain and calibration coefficient $b = 0.522$ depends on the trigger geometry and must be around 0.5 because the scattered intensity is proportional to the geometric cross section of the

particle (Vochezer et al., 2016). For each measurement campaign, the factor $a$ is calibrated with spherical droplets where the size is determined comparing the Mie fringes of the diffraction patterns to Mie theory (see Vochezer et al. (2016) for details). The ice crystal maximum dimension is estimated to be approximately 1.0 to 2.5 times larger than the spherical equivalent diameter depending on the crystal shape and complexity (see appendix A).

Besides particle size, other microphysical features can be extracted from the diffraction patterns recorded by PPD-2K and

SID-3, such as particle shape and degree of crystal complexity. The degree of crystal complexity can be derived from a speckle pattern texture analysis of the diffraction patterns, and is represented by the ice crystal normalised energy feature parameter $k_e$ (Schnaiter et al., 2016). $k_e$ is a measure for particle complexity with scales around the laser wavelength of the incident light, 532 nm, including surface roughness, polycrystallinity and (stepped) hollowness (Järvinen et al., 2023). The parameter ranges

between about 3.8 and 7.0 and a threshold of $k_{\mathrm{e}}^{\mathrm{thr}} = 4.6$ was defined by Schnaiter et al. (2016) for the SID-3 measurements, where lower values correspond to pristine particles and higher values for morphologically complex particles. Similarly to Schnaiter et al. (2016), $k_{\mathrm{e}}$ is only calculated for particles with trigger intensities between 10 counts and 25 counts in this work to reduce the effect of particle size biases on $k_{\mathrm{e}}$, which is caused by varying mean intensity of the diffraction pattern images. In this work, the probability of coincident particle sampling is below 1 % with a maximum detected particle concentration of $42.6\,\mathrm{cm}^{-3}$ and $22.3\,\mathrm{cm}^{-3}$ measured by PPD-2K and SID-3 (Vochezer et al., 2016).

Furthermore, the particle shape is derived from the diffraction patterns. If there are clear maxima in the azimuthally integrated polar profile (Mie fringes) the particle is considered spherical and is classified as a droplet (see Fig. 2d). Otherwise, a discrete fast Fourier transform (FFT) of the polar integrated azimuth intensity profile is performed for the shape identification following the procedure of Vochezer et al. (2016). Maximum Fourier coefficients of order 2 and 4 indicate a columnar shape (Fig. 2c) and maximum Fourier coefficients of order 3 and 6 indicate a hexagonal plate (Fig. 2a). If the maximum coefficient is of a non-symmetric order the particles are interpreted as irregulars (Fig. 2b). It needs to be noted that if a particle is classified as irregular it does not necessarily mean that it has an irregular shape. Irregular diffraction patterns may also result from sufficiently roughed columnar or hexagonal ice particles, which do not show their usual, distinct diffraction patterns. Furthermore, the fraction of columns and plates should be seen as a lower limit because the random particle orientations can lead to bent arcs in the scattering patterns of pristine hexagonal ice particles, which may occasionally be falsely interpreted as irregulars. Out of 100 random ice particles classified as irregulars by the Fourier method, 11 hexagonal ice particles with bent arcs were manually identified.

In this work, the measured single particle scattering information is converted into particle size distributions using 50 bins in a size range between about $7\,\mu\mathrm{m}$ and $70\,\mu\mathrm{m}$ depending on the campaign-specific size calibration. The particle size distributions are averaged over $10\,\mathrm{s}$ and a log-normal size distribution function is fitted to obtain the geometric mean diameter ($GMD$) and the standard deviation of the logarithm of $GMD$ ($\sigma$). The log-normal particle size distribution is defined as (Feingold and Levin, 1986; Tian et al., 2010):

$$n(d) = \frac{n_0}{\sqrt{2\pi} \cdot d \cdot \log \sigma} \cdot e^{\left(-\frac{\log^2 \frac{d}{GMD}}{2\log^2 \sigma}\right)} \tag{3}$$

where $n_0$ is the total concentration and $d$ is the particle diameter. Fig. 2e shows the retrieved log-normal parameters $\sigma$ over $GMD$. Most $\sigma$ are in a range between 1.1 and 1.5 with some outliers to higher values. $\sigma$ has a mean value of $1.26 \pm 0.13$. For all curve fits with $GMD < 10\,\mu\mathrm{m}$, $\sigma$ has a mean value of $1.21 \pm 0.08$.

To analyse the size of ice crystals that are smaller than the lower size limit of PPD-2K ($7\,\mu\mathrm{m}$) a Fourier transform infrared spectrometer (FTIR) was used to determine the particle size during the HALO06 campaign (Wagner et al., 2006). The FTIR measures the spectral extinction of the ice particle ensemble at wave numbers between $6000\,\mathrm{cm}^{-1}$ and $800\,\mathrm{cm}^{-1}$. At mid-infrared wavelengths, the extinction spectra only vary slightly with particle shape unless highly irregular habits are involved. For this study, similar to Schnaiter et al. (2012), the ice particle size distribution was assumed to be log-normal and retrieved from the measured extinction spectra using T-matrix calculations for a fixed aspect ratio of 0.7 (circular columns). The FTIR retrieval provides reliable results down to mean particle maximum dimensions of about 1 micron.

Furthermore, a formvar replicator was operated in the cloud chamber during measurement campaigns RICE01 and RICE03 to generate replicas of the ice crystals on a 35 mm transparent plastic film strip. Details can be found in Schnaiter et al. (2016). The formvar replicas are analysed using a Zeiss IM35 inverted microscope with a magnification of up to 500 and an Imaging Source DFK41AU02 camera with a resolution of 1280 x 960 pixels. The images of the replicas allow to obtain additional information about the particle shape that cannot be derived from the diffraction patterns, such as information about hollowness, but the statistics are poorer.

## 2.3 Experiment procedure

The expansion experiments were conducted in the Aerosol Interactions and Dynamics in the Atmosphere (AIDA) cloud chamber at the Karlsruhe Institute of Technology (Möhler et al., 2005). Here, the data from different ice nucleation measurement campaigns at cirrus temperatures between -75 °C and -39 °C are analysed (see Table 1). The temporal evolution of a typical experiment is shown in Fig. 3. The expansion experiments are conducted with the following procedure discussed in Schnaiter et al. (2016):

1. Preparation: The cloud chamber is cleaned by evacuation and flushing cycles and then humidified to ice-saturated conditions by forming a thin ice coating on the inner chamber walls.

2. Aerosol addition: The aerosols are added with an aerosol generator. The particle types were soot or mineral dust for heterogeneous ice nucleation experiments and sulphuric acid solution droplets for homogeneous freezing experiments.

3. Initial cloud activation: A cloud chamber expansion is started by opening the valve to the vacuum pumps. The start of pumping is indicated in Fig. 3 as reference time zero. The pressure and gas temperature in the cloud chamber decrease (Fig. 3a) and the relative humidity ($RH$) increases (see Fig. 3b). $RH$ is measured with a tunable diode laser absorption spectrometer (Ebert et al., 2005). When the relative humidity with respect to ice ($RH_{ice}$) reaches homogeneous freezing conditions or exceeds the aerosol-specific threshold for heterogeneous ice nucleation, the ice nucleation process starts, e.g. at 2 min in Fig. 2d. The nucleated ice crystals deplete the supersaturation in the gas phase, leading to a reduction in $RH_{ice}$. Once the particles reach sizes that are larger than the detection limit of the PPD-2K instrument particles are detected.

4. Sublimation: In order to remove the ice crystal morphological complexity from the initial growth period, the expansion is stopped and dry synthetic air is fed into the chamber between $t = 5\,min$ and $t = 9\,min$ of the experiment, increasing the gas pressure again (Fig. 3a). This further reduces $RH_{ice}$ to below 100 % (Fig. 3b), leading to a decrease of the ice particle size by sublimation (Fig. 3d).

5. Regrowth: The dry air flow is stopped and the evacuation of the cloud chamber is resumed, leading to a second, controlled growth period at a defined $RH_{ice}$-level above 100 %. This can be seen between $t = 10\,min$ and $t = 15\,min$ (Fig. 3b), when the particle size increases again due to growth at this defined supersaturation (Fig. 3d). Multiple sublimation and controlled regrowth cycles can be done in one experiment.

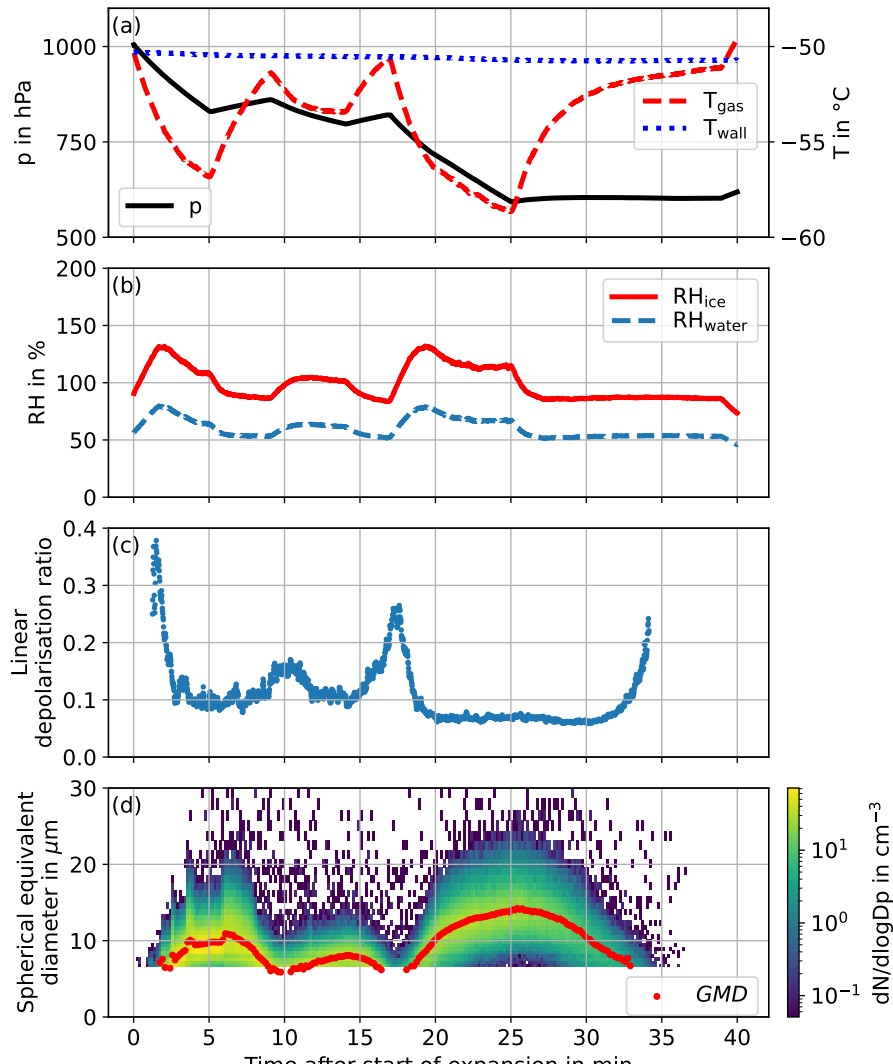

**Figure 3.** Example expansion experiment 06 from the RICE02 campaign. Initial growth, sublimation and two regrowth cycles can be seen. The gas pressure $p$, gas temperature $T_{\text{gas}}$ and wall temperature $T_{\text{wall}}$ inside the cloud chamber **(a)**, the relative humidity with respect to water $RH_{\text{water}}$ and ice $RH_{\text{ice}}$ **(b)**, the linear depolarisation ratio $\delta$ from SIMONE **(c)** and the geometric mean diameter ($GMD$) derived from from PPD-2K **(d)** are shown as a function of time after the start of the experiment.

6. Final sublimation: The expansion is stopped and due to the heat transfer from the chamber walls, whose temperatures have only slightly decreased during the expansion, the humidity in the cloud chamber falls below ice saturation, initiating the ice cloud sublimation. This can be seen in Fig. 3 for $t > 25$ min.

To correlate the linear depolarisation ratio to the particle size at cirrus temperatures, the following conditions are applied:

**Table 1.** Campaigns that are used for the analysis of the linear depolarisation ratio.

| Campaign | Time | SIMONE wavelength | Size data from | Number of experiments analysed |
|----------|------|-------------------|----------------|-------------------------------|
| HALO06 | Jan−Feb 2011 | 488 nm | FTIR | 11 |
| RICE01 | Nov 2012 | 488 nm | PPD-2K | 9 |
| RICE02 | Apr−May 2014 | 488 nm | PPD-2K | 14 |
| RICE03 | Dec 2014 | 552 nm | PPD-2K | 13 |

- The gas temperature of the cloud chamber $T_\mathrm{gas}$ is below -39 °C.

- The expansion was started.

- The SIMONE forward scattering intensity is above a threshold value to obtain high enough counts on the photomultipliers for an accurate retrieval of $\delta$ ($10^6$ counts for SIMONE and $10^4$ counts for SIMONE-Junior). The SIMONE instruments use an automated neutral density filter system to avoid saturation of the photomultiplier and increase the range of detection. The counts measured with the neutral density filter are normalized to the counts without neutral density filter.

- The $GMD$ determined by a log-normal fit is at most 20 % smaller than the center of smallest bin of the PPD-2K size range to ensure an accurate retrieval of $GMD$: $1.2 \cdot d_\mathrm{min} \leq GMD$. The center of the smallest bin ranges between 6.4 μm and 7.9 μm depending on the campaign specific calibration and photomultiplier gain settings.

- The particle concentration measured with PPD-2K is larger than a lower threshold of 0.3 cm$^{-3}$ to ensure PPD-2K measures enough particle events during the 10 s integration time for a log-normal fit to the particle size distribution.

Only those data points of the linear depolarisation ratio and $GMD$ are used for further analysis where the conditions above are fulfilled. This leads to 24 % of data being discarded due to PPD-2K limitations. This discarded data is predominantly due to low particle concentration or small particle sizes that regularly occur in the sublimation phase of the expansion experiments and can only be detected by SIMONE and not by PPD-2K. The data is averaged over the same 10 s periods for both instruments.

## 2.4 Numerical simulations

Transition matrix (T-matrix) (Mishchenko and Travis, 1998) and conventional geometric optics (CGOM) ray tracing Monte Carlo (Macke, 2020; Macke et al., 1996b, a) methods are used to simulate the near-backscattering depolarisation ratio that is measured with the SIMONE instrument. Here we use the T-matrix code from Leinonen (2014) to perform the T-matrix simulations of spheroidal particles, which are constrained to small size parameters below approximately 50 due to computational limitations. The assumption of spheroidal particle shapes is a rough approximation of the hexagonal shape of ice particles (Bailey and Hallett, 2009). Therefore, as a comparison to the T-matrix approach, which is widely used in LIDAR applications (e.g. Nicolae et al., 2018; Veselovskii et al., 2010; Mishchenko et al., 2000), light scattering simulations of hexagonal ice crys-

tals with the Invariant-Imbedding T-matrix Method (IITM) for size parameters of up to 158 are additionally compared to the measurement data.

For size parameters larger than 90 we use CGOM simulations of hexagonal particles with a tilted facet method and internal scatterers to generate complex ice crystals. The size parameter is defined as $x = \frac{2\pi a}{\lambda}$ with characteristic particle length $a$ and wavelength $\lambda$ of the light in the surrounding medium. A wavelength of $\lambda = 488$ nm and a refractive index of $n = 1.31$ are used.

All simulations calculate the scattering matrix elements that are needed to obtain the linear depolarisation ratio. For this, the assumption of randomly oriented ice particles is used. The linear depolarisation ratio for incident polarisation parallel to the scattering plane $\delta$ for an arbitrary scattering angle $\theta$ is defined as (Mishchenko and Hovenier, 1995):

$$\delta(\theta) = \frac{S_{11}(\theta) - S_{22}(\theta)}{S_{11}(\theta) + 2S_{12}(\theta) + S_{22}(\theta)} \tag{4}$$

where $S_{11}(\theta)$, $S_{12}(\theta)$ and $S_{22}(\theta)$ are the scattering matrix elements obtained from the CGOM and T-matrix simulations for the scattering angle $\theta$.

The T-matrix method is applied to simulate $\delta$ for the light scattered by randomly oriented pristine spheroids with aspect ratios ranging between 1.5 and 2.0. This range is based on the replica images that are presented later. In order to be comparable to the measurements, the T-matrix scattering matrix elements are integrated over a log-normal size distribution similar to Saito et al. (2021) with 82 particle sizes varying from 0.01 μm to 10.0 μm and a fixed $\sigma$ of 1.21. This is the mean $\sigma$ found in the log-normal fits to the PPD-2K particle size distributions with $GMD$ smaller than 10 μm. The maximum dimension of the spheroids is used for the comparison to the PPD-2K spherical equivalent diameter.

Furthermore, we use a dataset of numerically exact IITM simulations of hexagonal ice crystals published in Saito and Yang (2023). It simulates particle complexity as surface roughness. The degree of this surface roughness is defined by the variance ($\sigma^2$) of a two-dimensional Gaussian distribution of local planar surface slopes. The scattering matrix elements are integrated over a log-normal size distribution with 31 particle sizes between 0.1 μm to 24.6 μm, using a fixed $\sigma$ of 1.21 for particle sizes of up to 10 μm and a fixed $\sigma$ of 1.26 for larger particle sizes. The dataset was computed using the refractive index of ice at 532 nm ($n = 1.3116 - 1.49 \cdot 10^{-9} i$), which differs slightly from that at 488 nm. This difference has a marginal effect on light scattering in the backward direction. The aspect ratio ranges from 1.0 to 2.0, with decreasing maximum particle size for increasing aspect ratio due to increasing computational effort.

The CGOM simulations use hexagonal ice particles and average over 2 million particle orientations. The calculated scattering matrix elements are integrated over a log-normal size distribution with 42 particle sizes between 8.4 μm to 37.3 μm and a fixed $\sigma$ of 1.26. This is the mean $\sigma$ of all fits to PPD-2K cirrus particle size distributions. Ice crystal complexity is represented with the tilted facet approach where a random tilt angle is applied to the ice crystal surface. A distortion parameter describes the maximum possible angle of a random tilt that is applied to the crystal surface. With a distortion of 0 no tilt is applied and a distortion of 0.5 is equivalent to a random tilt of up to 45°. Additional ice crystal complexity can be simulated with the mean free path by adding internal scatterers to the simulation that randomly change the direction of the internal rays (Macke et al., 1996b). In this work, the internal scattering is non-absorbing with a single scattering albedo of 0.999, representative of air bubbles. The mean free path is the mean distance that a simulated ray propagates through the crystal in the Monte Carlo

simulation before changing direction due to the simulated internal scatterer. We have no direct information about concentration, size and type of internal scatterers inside the cloud chamber grown ice particles. Therefore, the mean free path is varied between 150 μm for high internal scattering and $10^4$ μm for negligible internal scattering to find the best overlap with the measurement data. The column length is used for the comparison to the PPD-2K spherical equivalent diameter.

The conversion between the particle sizes from the numerical simulations and the spherical equivalent diameter measured with PPD-2K is non-trivial and therefore an approximation is needed. In this work, we use the maximum dimension of the spheroidal particles and the column length of the hexagonal particles, which is common approach when analysing the optical properties of ice particles (Liu et al., 2014). To estimate possible errors, the spherical equivalent diameter measured with PPD-2K is calculated for different complex shaped ice particles in Fig. A1. The maximum particle dimension can be up to 40 % larger than the spherical equivalent diameter for rough columnar and bullet rosette shaped ice particles. However, the column length of columnar particles can also be smaller than the maximum particle dimension, depending on its aspect ratio (Um et al., 2015). For example, a columnar particle with an aspect ratio of 1.5 has a maximum dimension, which is about 30 % larger than its length. These uncertainties in size conversion are shaded in the figures where the results of the numerical simulations are compared to the measurements with PPD-2K.

## 3   Results

Since the linear depolarisation ratio is sensitive to the particle morphology, we start by giving an overview of the microphysical properties of our laboratory generated ice crystals. The key microphysical properties that were recorded were particle size and small-scale complexity. Then, the observed linear depolarisation ratio is shown as a function of particle size and small scale complexity and the observational results are compared to numerical simulations.

### 3.1   Microphysical properties of the cloud chamber grown cirrus

PPD-2K measured particle spherical equivalent diameters of up to 70.5 μm for ice crystals nucleated and grown in the AIDA chamber at gas temperatures below -39 °C. Fig. 4 shows histograms of the $GMD$ of ice crystals grown between AIDA initial gas temperatures between -45 °C and -39 °C and between -75 °C and -45 °C, respectively. At initial gas temperatures between -45 °C and -39 °C, the median $GMD$ was 17.4 μm (Interquartile range ($IQR$) = 6.7 μm). In comparison, ice crystals grown at lower cirrus temperatures between -75 °C and -45 °C had a median $GMD$ of 10.6 μm ($IQR = 3.6$ μm). The $GMD$ of ice crystals grown in the lower cirrus temperature range is 39 % smaller than of ice crystals grown in the higher cirrus temperature range. It is expected that ice crystals at higher temperatures grow to larger sizes, since at the same relative humidity the growth rate increases for increasing temperatures due to more available condensible water vapour (Bailey and Hallett, 2009). Ice crystals grown in the AIDA cloud chamber were limited to maximum sizes of about 70 μm due to sedimentation losses.

A visual analysis of the ice crystal shapes was performed using microscope images of the formvar replicas during regrowth phases when the relative humidity was kept constant. This is done for a subsample of the cloud chamber experiments (see Table 2). The aim is to identify differences in the growth characteristics on the formvar replicas at different growth conditions.

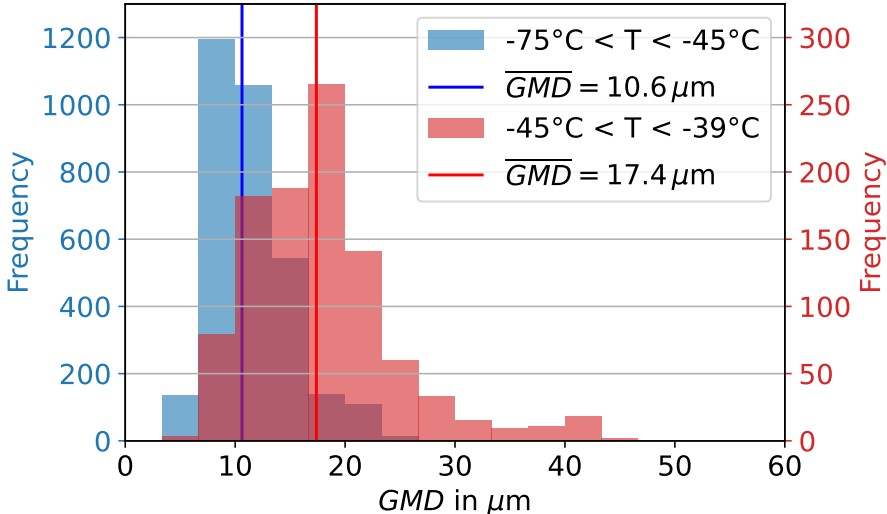

**Figure 4.** Histograms of geometric mean diamaters ($GMD$) derived from 10 s averaged PPD-2K particle size distributions for initial AIDA gas temperatures between -75 °C and -45 °C (blue) and between -45 °C and -39 °C (red). The vertical lines show the median values for the distributions. The median $GMD$ of ice crystals grown at lower cirrus temperatures is 39 % smaller than of ice crystals grown at higher cirrus temperatures.

The relative humidity with respect to ice ($RH_{ice}$) in the regrowth phases is varied between 105 % and 120 %. The number of
305 columnar ice crystals and the number of hollow columnar ice crystals were manually counted per microscopic frame, which usually contained some tens of ice crystals. Out of the 11096 imaged ice crystals on 324 microscope frames. 29 % were classified having columnar and 71 % of ice crystals were classified having other, mainly irregular shapes (see exemplary replicator images in Fig. 5a-e). The irregular shapes were predominantly compact crystals that sometimes showed several c-axes radiating from a center point, resembling budding rosettes (e.g. Fig. 5d). The columnar growth regime at temperatures below -40 °C is
310 consistent with previous laboratory studies, e.g. by Bailey and Hallett (2009). They reported that for relative humidities below $RH_{ice} = 125\%$ mainly single columns grow, while for higher relative humidity (budding) rosettes are expected. In Table 2 the fraction of replicas classified as columnar is shown for different relative humidity with respect to ice between 105 % and 120 % and for different gas temperatures. At -40 °C the column fraction ranges between 12 % and 25 % with no clear dependence on the relative humidity with respect to ice. At -50 °C the fraction of replicas classified as columnar is larger than at -40 °C,
ranging from 23 % ($RH_{ice} = 105\%$) to 59 % ($RH_{ice} = 120\%$). At this temperature range the columnar fraction increases with increasing supersaturation in the regrowth phase. The higher column fraction with a mean of 41 % at -50 °C in comparison to a mean of 19 % at -40 °C agrees well with previous results of laboratory-grown ice crystals by Bailey and Hallett (2009), where at -50 °C more single crystal columns are expected in comparison to the transition region between columnar and plate-like growth regimes at -40 °C. The observed transition towards columnar growth with increasing relative humidity between -40 °C
and -50 °C for $RH_{ice} < 130\%$ has also been reported. Example microscope images of formvar replicas from experiment 29,

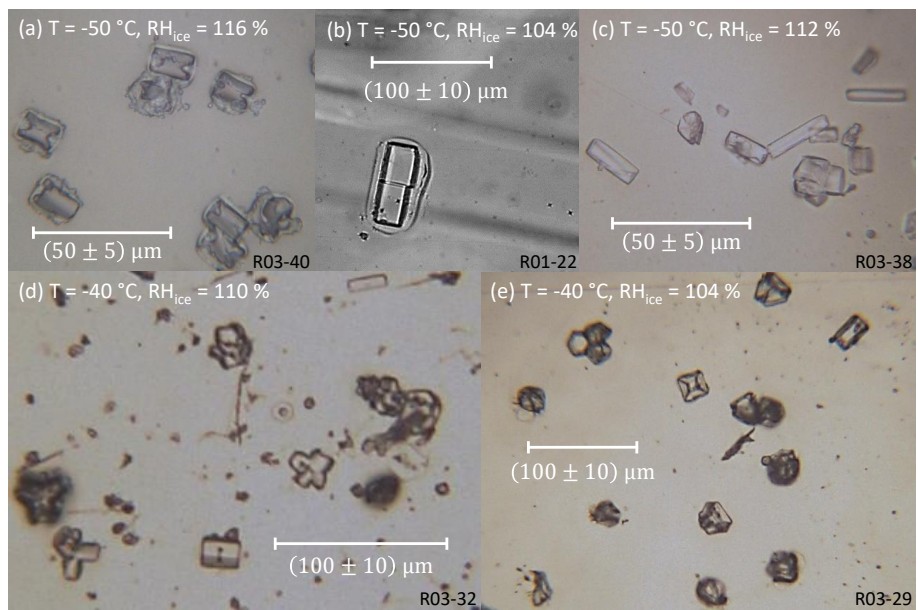

**Figure 5.** Microscopic images of example formvar replicas of cloud chamber grown ice crystals highlighting different ice crystal complexities. The ice particles were grown at temperatures of -40 °C and -50 °C and relative humidity between 104 % and 116 %. R03-30 stands for experiment 30 of measurement campaign RICE03. The hexagonal ice crystals in **(a)** show hollowness on the basal facets and occasionally have air inclusions and the ice crystal in **(b)** has a central dislocation. **(c)** and **(e)** show columnar and irregularly shaped ice crystals and in **(d)** budding rosettes can be seen.

32, 38 and 40 of the RICE04 campaign are shown in Fig. 5e,d,c,a. Due to the small sample number, the fraction of columnar particles and hollow columnar particles seen in Fig. 5 is not representative. Nonetheless, the general occurrence of hollowness on the prism faces at both cirrus temperatures and the smaller particles sizes at -50 °C in comparison to -40 °C can be observed on the microscope images.

Statistically more robust information about ice crystal shapes can be derived from the SID-3 diffraction patterns using the Fourier analysis method. Of all particle images at cirrus temperatures of the investigated campaigns 0.4 % show diffraction patterns with features of spherical particles, 3.8 % show diffraction patterns with features of plates, 35 % show diffraction patterns with features of columns and 61 % show diffraction patterns with features of irregular ice particles. This is in good agreement with the fractions of columnar and irregular particles identified on the microscope images of the replicas taken during the regrowth phases of a subsample of the experiments (see Table 2). In Fig. 6a the fractions of the ice crystal shapes are shown for the different campaigns and temperature groups. It can be noted that the fraction of columns increases from 19 % (RICE01) and 18 % (RICE03) for initial gas temperatures between -45 °C and -39 °C to 44 % (RICE01), 36 % (RICE02) and 38 % (RICE03) between -75 °C and -45 °C. This characteristic was also seen in the replica analysis.

Furthermore, different types of morphological complexities can be studied with the ice crystal replicas. The most common type of complexity is hollowness of the basal facets (Fig. 5a) with occasional air inclusions (Fig. 5e). Table 2 shows the fraction

**Table 2.** Results of the analysis of the microscope images of the formvar replicas taken during regrowth phases. Only experiments are analysed where a stable relative humidity during the regrowth phase was achieved. $N_{\text{tot}}$ is the total number of investigated ice particles. $f_{\text{col}}$ is the fraction of identified columnar particles of the total number of particles and $f_{\text{hollow}}$ is the fraction of identified hollow columnar particles of all identified columnar particles. $f_{\text{col}}$ (SID-3) is the fraction of columnar particles derived from the SID-3 diffraction patterns during the regrowth phases of the experiments as a comparison.

| $T_{\text{gas}}$ in °C | -40 | | | | -50 | | | | |
|---|---|---|---|---|---|---|---|---|---|
| Campaign | RICE01 | RICE03 | | | RICE01 | RICE03 | | | |
| Experiment | 27 | 29 | 30 | 32 | 20 | 38 | 39 | 40 | 41 |
| $N_{\text{tot}}$ | 508 | 787 | 764 | 1516 | 424 | 1350 | 1113 | 638 | 406 |
| $f_{\text{col}}$ | 25 % | 22 % | 12 % | 17 % | 23 % | 40 % | 39 % | 42 % | 59 % |
| $f_{\text{col}}$ (SID-3) | 42 % | 24 % | 23 % | 19 % | 39 % | 25 % | 43 % | 43 % | 47 % |
| $f_{\text{hollow}}$ | 45 % | 33 % | 73 % | 36 % | 12 % | 4 % | 36 % | 70 % | 29 % |
| $RH_{\text{ice}}$ mean | 120 % | 105 % | 107 % | 111 % | 105 % | 109 % | 108 % | 112 % | 120 % |

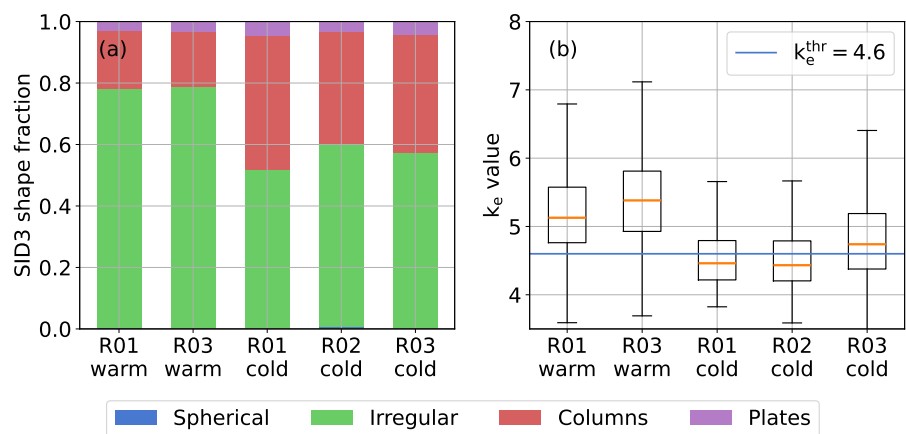

**Figure 6.** Particle shape fractions that are determined from all forward scattering images taken with the SID-3 instrument during the different campaigns **(a)**. Each campaign is divided into two different groups. Cold refers to AIDA initial gas temperatures between -75 °C and -45 °C and warm to temperatures between -45 °C and -39 °C. In **(b)** $k_{\text{e}}$ is plotted over the different campaigns. $k_{\text{e}}^{\text{thr}}$ is the threshold of 4.6 for morphologically complex ice crystals defined by Schnaiter et al. (2016). More complex ice particles are observed for the higher cirrus temperatures.

of hollow columns to all columns. This fraction varied between 33 % ($RH_{\text{ice}} = 105\,\%$) and 73 % ($RH_{\text{ice}} = 107\,\%$), with a mean of 46 % for experiments initiated at -40 °C. At initial gas temperatures of -50 °C the mean fraction of columns that are hollow per experiment ranges between 4 % ($RH_{\text{ice}} = 109\,\%$) and 70 % ($RH_{\text{ice}} = 112\,\%$), with a mean of 40 %. Hollow fractions below 30 % are only observed for experiments at initial gas temperatures of -50° C. Thus, for both investigated temperatures a significant mean fraction of columns (more than 40 %) show hollowness. This is in contrast to a laboratory study by Harrington

and Pokrifka (2024), who estimated a critical supersaturation of 20 % for columns to develop hollowness at the basal facets at temperatures below -40 °C. It is not always possible to visually identify hollowness on the microscope images of the formvar replicas. Therefore, the observed fraction of columnar ice particles and hollow columnar ice particles should be considered as a lower limit. In atmospheric cirrus observations Walden et al. (2003) found all bullets but only few columns to be hollow in microscope pictures of precipitation at cirrus temperatures at South Pole station between -73 °C and -35 °C. Schmitt and Heymsfield (2007) observed hollow ends in 52 % to 80 % of all bullet-rosette and columnar particles of formvar replicas from balloon-borne mid-latitude cirrus observations between -46 °C and -33 °C. The increasing trend in the fraction of columns with hollow ends with temperature is in agreement with our cloud chamber observations. Another type of crystal complexity that was observed on the replicas are dislocations (Fig. 5b). They are thought to form from thermal stress in the changing growth conditions that can be present during the experiments (Baker, 2003).

Due to the optical limitation of the microscope and the replication technique, it is not possible to make conclusions about sub-micron scale complexity, such as surface roughness, from the microscope replica analysis. For this, we use the crystal complexity information derived from the SID-3 diffraction patterns. Fig. 6b shows a statistical analysis of the optical complexity parameter $k_e$ for the different campaigns and temperature groups. For the ice crystals grown in the higher temperature range, the median $k_e$ is with 5.22 clearly higher than the threshold of 4.6 defined for morphologically complex ice crystals. In the lower cirrus temperature range median $k_e$ is with 4.54 below the threshold of complex ice crystals. One reason can be that higher growth rates at higher temperatures promote crystal complexity, as discussed in Schnaiter et al. (2016). In addition, small differences can be seen in the fraction of columnar ice crystals and $k_e$ for the different measurement campaigns in the same temperature groups. For example, RICE03 has a higher median $k_e$ in comparison to RICE02 and RICE01 in both temperature groups. These differences can be explained by different $RH_{ice}$ during the experiments of the different campaigns. In the next section, we analyse the depolarisation properties for each of the two temperature ranges separately.

### 3.2 Effects of particle size on the linear depolarisation ratio

In Fig. 7 the measured $\delta$ is shown as a function of $GMD$ derived from the PPD-2K particle size distributions for the previously defined temperature ranges. Fig. 7a shows the data in the higher temperature range between -45 °C and -39 °C, where $\delta$ has a minimum of 0.08 for a $GMD$ around 12 µm and increases for both larger and smaller $GMD$ up to 0.3. In the lower temperature range between -75 °C and -45 °C during RICE01 and RICE02, $\delta$ has a constant low value between 0.05 and 0.10 for sizes between 8 µm and 25 µm (Fig. 7b). For smaller sizes below 8 µm observed during the HALO06 campaign, $\delta$ increases sharply and reaches values up to 0.5. These are two distinct temperature-dependent depolarisation ratio - size modes. However, the data from the RICE03 campaign show a similar behaviour as for the higher temperature range in Fig. 7a. This difference is likely caused by more complex crystals during the lower cirrus temperatures of RICE03 with a median higher than the threshold value of $k_e^{thr} = 4.6$ (see Fig. 6). A detailed comparison to previous atmospheric observations is given in section 4. In the following section, we investigate the link between $\delta$ and the optical crystal complexity.

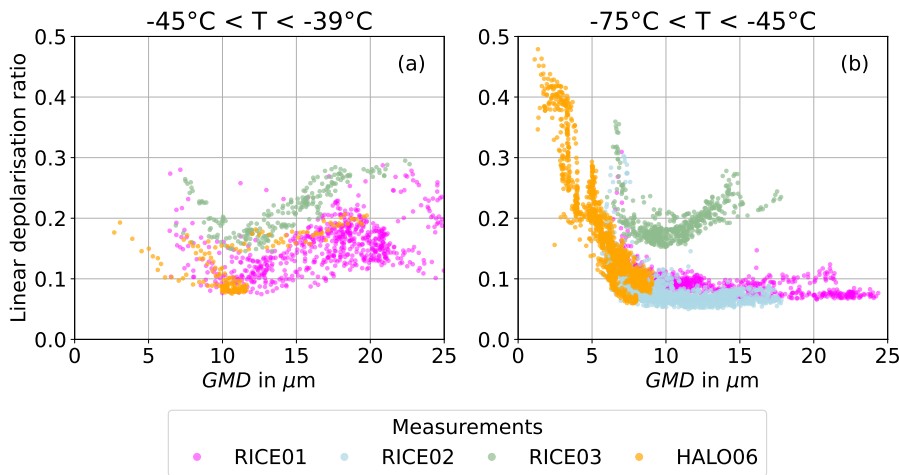

**Figure 7.** Linear depolarisation ratio measured with SIMONE as a function of geometric mean diameter ($GMD$) of the ice particles measured with the PPD-2K and FTIR instruments. **(a)** includes all measurements with initial cloud chamber temperatures between -45 °C and -39 °C and **(b)** between -75 °C and -45 °C. Each data point represents 10 s averaged measurement data out of multiple experiments, which consisted of multiple growth and sublimation phases with different pump speeds.

### 3.3 Effects of the optical crystal complexity on the linear depolarisation ratio

In Fig. 8 the measured $\delta$ is shown as a function of the optical crystal complexity metric derived from SID-3 measurements
for all available cirrus temperatures between $-75°C < T < -39°C$. Pearson product-moment correlation coefficients $R$ are determined and a linear regression is added for $R > 0.5$. This is done separately for each campaign where SID-3 was operated. The results are separated into size groups of $(5\pm1)\,\mu m$, $(10\pm1)\,\mu m$, $(15\pm1)\,\mu m$ and $(20\pm1)\,\mu m$ to limit the effect of a possible size dependence of $k_e$.

For the smaller three particle size ranges of $(5\pm1)\,\mu m$, $(10\pm1)\,\mu m$ and $(15\pm1)\,\mu m$ $|R| \leq 0.13$ indicates no or weak cor-
relation between $k_e$ and $\delta$. Only for the largest size range of $(20\pm1)\,\mu m$ a moderate correlation of $R = 0.54$ is observed. The scarcity of particles in this largest size group during the RICE02 measurement campaign can be attributed to the lower initial gas temperature of $-50°C$, compared to RICE01 and RICE03, which included experiments at $-50°C$ and $-40°C$. Higher growth temperatures promote the formation of larger ice particles through increased growth rates at the same relative humidity (Bailey and Hallett, 2009).

The weak to moderate correlation between the optical complexity parameter $k_e$ and the depolarisation ratio $\delta$ contrasts with previous studies, which have reported increasing $\delta$ with increasing surface roughness in the 5 - 20 µm size range (Smith et al., 2016; Saito and Yang, 2023). Using numerical simulations, Saito and Yang (2023) showed that $\delta$ increases with increasing surface roughness up to a threshold of $\sigma^2 \approx 0.1$, beyond which additional roughening has little to no further effect on $\delta$. Most of the crystals in our experiments fall within the range $k_e > 4.6$, which could indicate that their surface roughness was
already above this threshold—potentially explaining the weak dependence of $\delta$ on optical complexity. Schnaiter et al. (2016)

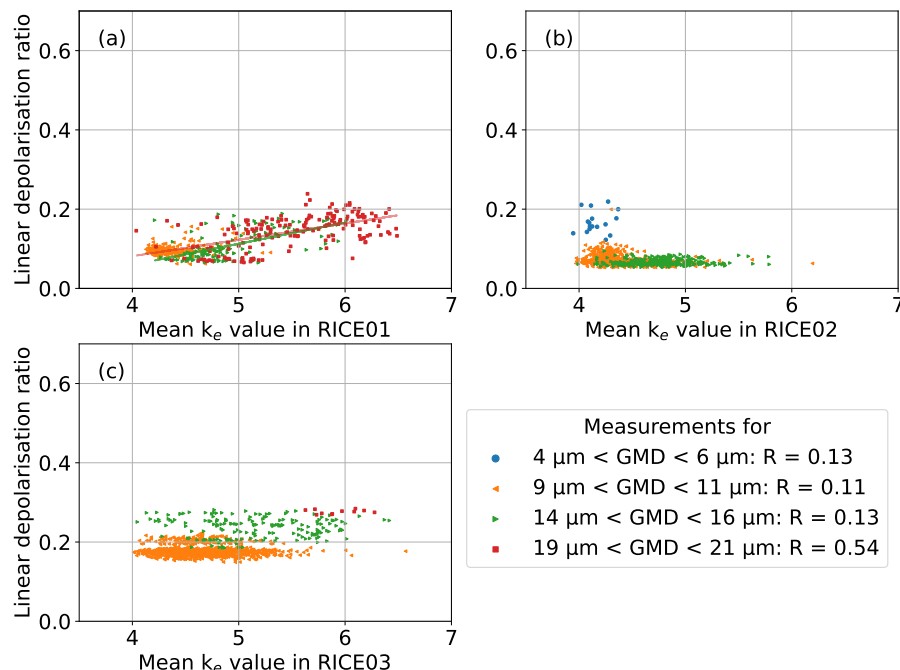

**Figure 8.** Linear depolarisation ratio as a function of optical particle complexity $k_e$ measured with the SID-3 instrument for the RICE01 campaign **(a)**, for the RICE02 campaign **(b)** and for the RICE03 campaign **(c)** with linear fits to the data. Only a weak to moderate correlation is observed.

found that $k_e > 4.6$ is indicative of moderately to strongly roughened crystals, supporting this interpretation. Growing more pristine crystals with $\sigma^2 < 0.1$ would require very low supersaturations and long growth times, conditions that are difficult to maintain in the AIDA cloud chamber due to sedimentation losses. Alternatively, other morphological features not captured by $k_e$, such as crystal habits, may have varied across experiments and influenced $\delta$, further weakening the observed correlation. This highlights the complexity of the relationship between $\delta$ and ice crystal morphology.

## 3.4 Comparison of measurements to numerical simulations

In this section, the measurement data for the two temperature ranges are compared to two types of T-matrix simulations and to conventional geometric optics numerical (CGOM) simulations at the near-backscattering direction of 178°. Fig. 9 shows $\delta$ from T-matrix numerical simulations of spheroidal particles and CGOM simulations of columns with hollow basal facets, together with the measurement data from Fig. 7. The simulation parameters are detailed in section 2.4. With the assumption of spheroidal particle shapes—a rough approximation of the real, much more complex ice particle shapes—the T-matrix simulations reproduce the size-dependence of the measurement data well in the size range of about 2 μm to 9 μm. Overall, the T-matrix results of spheroidal particles are only meaningful in the lower temperature range (Fig. 9b), as for the warmer temper-

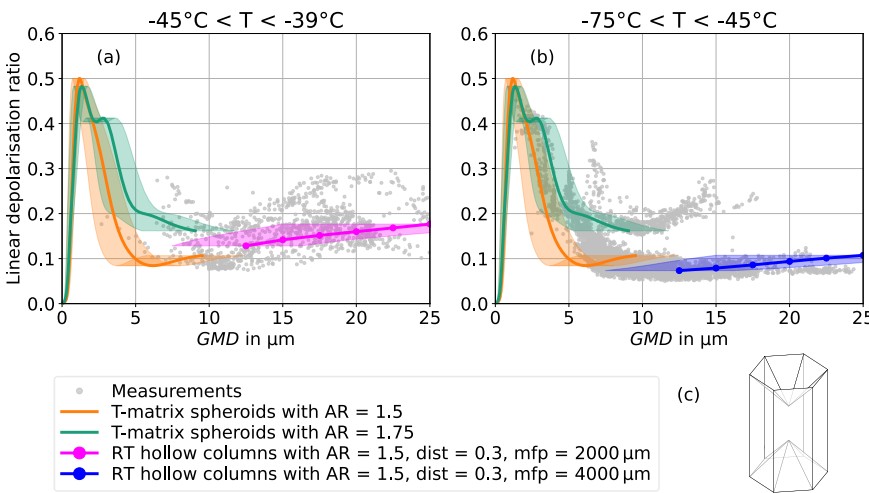

**Figure 9.** Linear depolarisation ratio as a function of geometric mean diameter ($GMD$). The measurement data (dots) are compared to T-matrix simulations of spheroidal particles at 178° near-backscattering direction (lines) (Leinonen, 2014) and to CGOM simulations of columns with hollow basal facets (Macke, 2020). The experimental data at initial gas temperatures between -45 °C and -39 °C is shown in **(a)** and between -75 °C and -45 °C in **(b)**. **(c)** shows the schematics of the used column shape with hollow basal facets entering 33.3 % of the column height from each prism facet. The shaded areas mark the uncertainty in size conversion between the measurement data and the simulations. There is a good overlap between the measurement data and the simulations.

ature range the ice crystals are too large to compute the light scattering properties with the method of (Mishchenko and Travis,
1998; Leinonen, 2014).

The CGOM simulations of $\delta$ use hollow columns where each basal facet is hollowed to a depth of 33.3 % of the length of the column. The schematics are shown in Fig. 9c. This type and length of hollowness is consistent with the ice crystal replicas that were found on the microscope images of the formvar slides (e.g. see Fig. 5a) and with the hollowness parametrisation of Zhu et al. (2023). The consideration of hollowness lowers the linear depolarisation ratios to the range of the measurement
data, likely due to the presence of additional planar surfaces inside the crystal (Yang et al., 2008; Smith et al., 2016). Using a mean free path around 2000 μm (**B**), the $\delta$ values of the higher cirrus temperature range between -45 °C and -39 °C can be reproduced. Using a mean free path of about 4000 μm (**A**), the CGOM simulations reproduce the $\delta$ values of the lower cirrus temperature range between -75 °C and -45 °C. The implementation of ice crystal complexity with hollowness, surface roughness and internal scatterers in the CGOM ray tracing methods allows to reproduce the measured $\delta$. CGOM simulations of
solid columns overestimate $\delta$, as shown in appendix B1, where ice crystal complexity is varied with distortion parameters from 0 (pristine) to 0.5 (highly complex). The CGOM simulation results are almost identical for aspect ratios of 1.75 and 2.0, and for the wavelength of $\lambda = 552$ nm, which was used during the RICE03 campaign (see appendix B3 and B4). CGOM simulations of hollow columns using mean free path from $10^4$ μm, indicating negligible internal scattering, to 500 μm, indicating high internal scattering are shown in appendix B2.

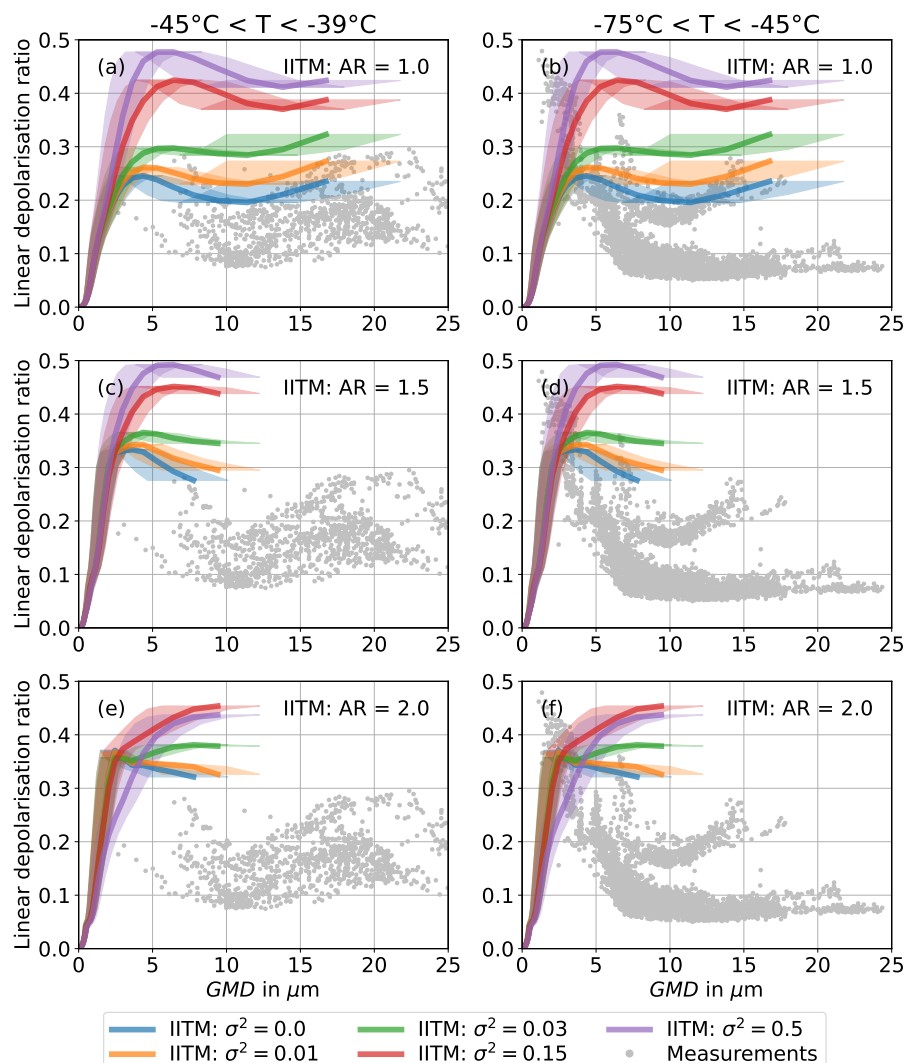

**Figure 10.** Comparison between the measured linear depolarisation ratio as a function of geometric mean diameter ($GMD$) (dots) and Invariant Imbedded T-Matrix (IITM) simulations of hexagonal particles at 178° near-backscattering direction (lines). Surface roughness variance $\sigma^2$ between 0 and 0.5 are used. The results for an aspect ratio of 1.0 are shown in **(a)** and **(b)**, for an aspect ratio of 1.5 in **(c)** and **(d)**, and for an aspect ratio of 2.0 in **(e)** and **(f)**. The experimental data at initial gas temperatures between -45 °C and -39 °C is added in **(a)**, **(c)** and **(e)**, and between -75 °C and -45 °C in **(b)**, **(d)** and **(f)** for comparison. The shaded areas mark the uncertainty in size conversion between the measurement data and the simulations. The IITM simulations without internal crystal complexity overestimate the measured $\delta$ for most sizes.

CGOM simulations use the geometric optics approximation and therefore only yield reliable results for size parameters larger than about $10^2$. Recent numerically exact T-matrix simulations have the capability to fill the size range between the smallest

particle sizes and ray tracing simulations in the geometric optics approximation for complex shaped hexagonal particles (Saito and Yang, 2023). Fig. 10 shows $\delta$ from IITM simulations with aspect ratios between 1.0 and 2.0. There is a sharp increase in $\delta$ with increasing particle size from $\delta \approx 0$ at particle sizes <1 μm to $0.2 < \delta < 0.5$ at particle sizes around 3 μm. For particle sizes larger than about 3 μm, $\delta$ reaches a plateau with smaller variation with size. The value of $\delta$ on the plateau increases with increasing surface roughness $\sigma^2$. The simulations with aspect ratios of 1.5 and 2.0 show slightly higher results for $\delta$ in comparison to simulations with an aspect ratio of 1.0. While the simulated $\delta$ is lower than the CGOM simulations of solid hexagonal particles (see appendix B1), it overlaps with the CGOM simulations of hollow columns at low mean free paths (see appendix B2). Yet, the $\delta$ from the IITM simulations still generally overestimates the measurement data. Only for the pristine case ($\sigma^2 = 0$) with aspect ratio 1.0, the simulations partly overlap with the measurement data. However, based on the microphysical data at least some degree of complexity on the ice crystal replica images and the optical small-scale complexity parameter is expected.

Similarities and differences in the evolution of $\delta$ with particle size from the IITM simulations are found in comparison to the T-matrix simulations. The sharp increase in $\delta$ for the smallest particle sizes in the IITM simulations is also seen in the T-matrix simulations. However, the peak of $\delta$ for the T-matrix simulations is at a lower size of about 2 μm in comparison to the IITM simulations. Furthermore, the subsequent plateau of $\delta$ is observed in both simulations, but at lower sizes and for lower $\delta$ in the T-matrix simulations in comparison to the IITM simulations. Further refining the ice crystal complexity model of the IITM simulations, for example, by incorporating hollowness, could enhance its agreement with the measurement data.

## 4 Discussion

In this section, the measurement results are compared to previous studies on the linear depolarisation ratio of cloud chamber grown and atmospheric cirrus clouds. Furthermore, the limitations of the numerical simulations are discussed.

Sassen and Benson (2001) observed a mean $\delta$ of 0.33±0.11 for cirrus with a mid-latitude ground-based lidar at a wavelength of 694 nm located at the facility for atmospheric remote sensing of the University of Utah. CALIPSO space-borne lidar measurements at 532 nm have given similar results for daytime measurements of cirrus with a global mean of $\delta = 0.34$, whereas the nighttime global average of $\delta = 0.24$ is closer to our cloud chamber findings (Sassen and Zhu, 2009). Yet, the authors explain the lower nighttime values as an artefact caused by background signals from Rayleigh scattering of the atmosphere. Most other lidar studies of mid-latitude cirrus reported $\delta$ in the range between approximately 0.2 and 0.5 (e.g. Wang et al., 2008; Kim et al., 2014; Urbanek et al., 2018; Manoj Kumar and Venkatramanan, 2020; Li and Groß, 2021; Gil-Díaz et al., 2024; Li and Groß, 2025). Polar cirrus are shown to have a lower $\delta$ between 0.1 and 0.3 compared to mid-latitude cirrus at the same temperature from ground-based lidar observations with a wavelength of 532 nm (Del Guasta, 2001; Del Guasta and Vallar, 2003) as well as from CALIPSO space-borne lidar measurements (Sassen et al., 2012). The cloud chamber grown ice particles that we observed in this study have $\delta$ between 0.08 and 0.3 at $GMD$ below 70 μm, which is in good agreement with polar studies. One potential explanation for lower $\delta$ values in polar observations is the common occurrence of diamond dust and ice fog particles at sizes smaller than 100 μm in high latitudes, which rarely exist in mid-latitudes (e.g. Walden et al., 2003; Gultepe et al., 2017).

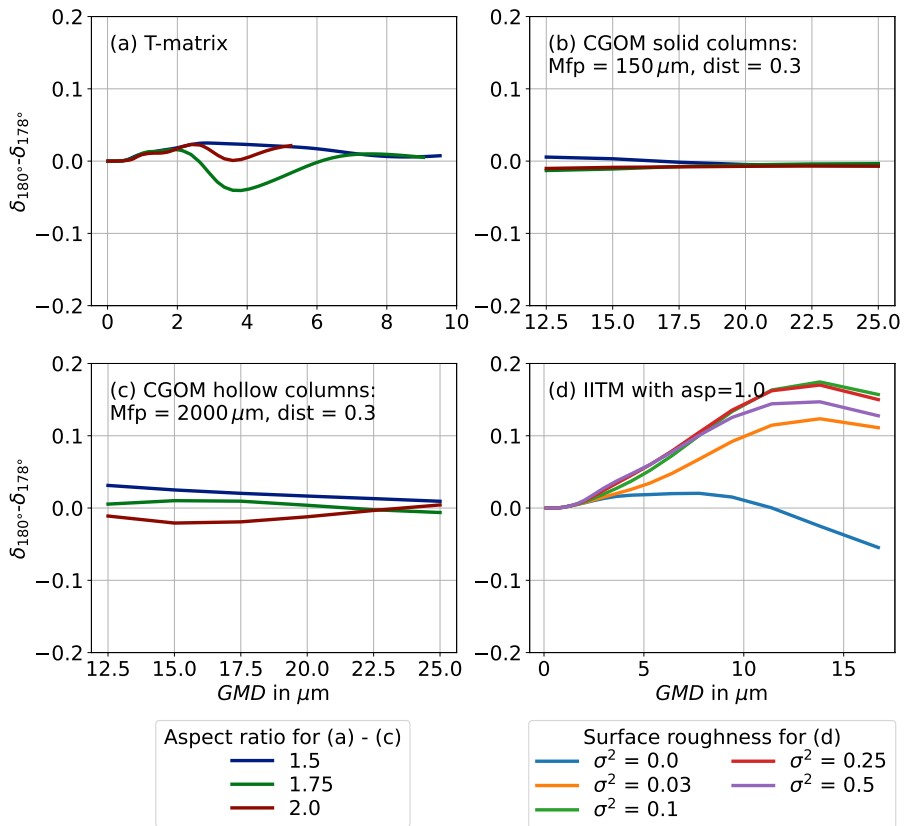

**Figure 11.** Difference between the linear depolarisation ratio at the exact backscattering direction ($\delta_{180°}$) and at 178° ($\delta_{178°}$) for T-matrix simulations **(a)**, CGOM simulations of solid columns with a distortion of 0.3 and a mean free path of 150 μm **(b)**, CGOM simulations of hollow columns with a distortion of 0.3 and a mean free path of 2000 μm **(c)** and IITM simulations with roughness variance $\sigma^2$ between 0 and 0.5 and an aspect ratio of one **(d)**. IITM simulations show the highest possible bias up to 0.17 for $GMD = 13.8$ μm.

Our laboratory results are restricted to relatively small ice crystals, whereas atmospheric ice crystals in cirrus regularly reach sizes larger than 100 μm, especially at higher cirrus temperatures (Mitchell et al., 2011; Woods et al., 2018; De La Torre Castro et al., 2023). Groß et al. (2023) investigated the linear depolarisation ratio of case studies of different cirrus between -63 °C and -58 °C during the CIRRUS in Mid-Latitudes (CIRRUS-ML) campaign. In the four investigated case studies, the linear depolarisation ratio also increases with increasing particle size from a median $\delta$ of 0.39 for the cloud with the smallest medium effective diameter of 52.2 μm to a median $\delta$ of 0.52 for the cirrus with the largest medium effective diameter of 193.8 μm. The

$\delta$ observed during CIRRUS-ML measured at the direct backscattering direction largely exceeded the $\delta$ observed at the AIDA cloud chamber at the near backscattering direction of 178°. The findings indicate that more comprehensive optical models are necessary to accurately reproduce the observed linear depolarisation ratios of morphologically complex ice crystals.

Small ice crystals with sizes below $10\,\mu m$ are known to cause high $\delta$ (Schnaiter et al., 2012). The cloud chamber grown ice clouds with $GMD$ below $10\,\mu m$ that we observed in this study showed $\delta$ of up to 0.45. This can explain the high $\delta$ observed in lower cirrus temperatures in the atmosphere (e.g. Sassen and Zhu, 2009; Li and Groß, 2021) by the presence of small ice crystals with spherical equivalent diameters below $5\,\mu m$. In situ observations of small ($1\,\mu m$<maximum dimension<$10\,\mu m$) ice crystals have been made in tropical tropopause cirrus at temperatures below -40 °C (Woods et al., 2018; Lawson et al., 2019). This can be a driver of higher $\delta$ observed for cirrus at lower latitudes where deep convective systems occur with higher frequencies (Futyan and Del Genio, 2007). Small ice particles in this size range also occur in contrails (Singh et al., 2024), influencing $\delta$ retrieved by lidar measurements.

Laboratory studies by Smith et al. (2016) investigated $\delta$ of artificially grown ice crystals at -30 °C with similar maximum dimensions between $20\,\mu m$ and $80\,\mu m$ at a wavelength of $532\,nm$. The ice crystals were grown in a fall tube and contained a higher fraction of plate-like particles according to the replica photographs. For the columnar particles hollowness on the basal facets and air inclusions were visible on the replica images, similar to our observations. The ice crystals were found to have $\delta$ measured at 180° backscattering direction between 0.25 and 0.45 and at 178° near-backscattering direction between 0.1 and 0.35, which is comparable to our measurements. This raises the question of how much lower our observed $\delta$ at 178° ($\delta_{178°}$) is in reference to exact backscattering $\delta$ at 180° ($\delta_{180°}$) (Schnaiter et al., 2012).

The simulated difference between $\delta_{180°}$ and $\delta_{178°}$ is shown in Fig. 11a for T-matrix simulations, in Fig. 11b for CGOM simulations of solid columns, in Fig. 11c for CGOM simulations of hollow columns and in Fig. 11d for IITM simulations. For T-matrix and CGOM simulations the largest difference $\delta_{180°} - \delta_{178°}$ is below 0.04, and thus cannot explain the difference between our cloud chamber observations and atmospheric values. Only the IITM simulations show a clear bias towards lower depolarisation ratios for the 178° measurements compared to measurements at the exact backscattering direction for roughened crystals (Fig.11d). This bias increases with increasing particle sizes, and for roughened crystals a maximum bias of 0.17 for a $GMD$ of $13.8\,\mu m$ is found. Therefore, it cannot be entirely ruled out that the slightly lower $\delta$ values observed in AIDA, compared to atmospheric lidar, are due to the deviation from exact backscattering. However, this offset does not affect the comparison with numerical models, which are also evaluated at the near-backscattering angle of 178°.

The results suggest that more comprehensive optical models are required to reproduce the observed depolarisation ratios of complex ice crystals. While the IITM simulations, which included only surface roughness on solid columns, showed poor agreement with measurements, the CGOM model achieved better correspondence only when multiple scales and types of complexity——such as hollowness and internal scatterers——were incorporated. Although the applicability of the CGOM simulations to correctly model polarimetric properties has been questioned, because changes in polarisation caused by internal scattering are omitted in the simulations, and interference effects from different rays leaving the particle are excluded (e.g. Smith et al., 2016; Macke et al., 1995), it was still possible to reproduce observed $\delta$ values when a suitable combination of complexity parameters was applied. This supports our conclusion that capturing the full range of morphological complexity is critical for accurate optical modelling. However, commonly used approaches like the tilted facet method remain limited, as they lack physical surfaces and are difficult to relate to real, observed ice crystal features (Macke et al., 1996a; Liu et al., 2013). Since many complexity features occur at scales comparable to the wavelength of light, their representation within geomet-

ric optics frameworks remains questionable. Together, these findings highlight the need for more physically realistic models that include both surface and internal complexity or even more complex shapes to accurately simulate polarimetric scattering properties.

## 5 Summary

In this study, we analysed the relationship between the linear depolarisation ratio and the simultaneously measured microphysical properties of laboratory-generated ice crystals, specifically size, shape and optical complexity. The ice crystals were grown under controlled cirrus conditions with sizes predominantly below $70\,\mu m$. The analysis of the microphysical properties of 47 ice clouds grown in the AIDA cloud chamber shows that smaller and more pristine ice crystals tend to form at lower cirrus temperatures between -75 °C and -45 °C, while larger and more complex crystals are more common at higher cirrus temperatures between -45 °C and -39 °C. According to formvar replica photographs, the fraction of hexagonal columnar particles increases with decreasing temperature from 19 % at -40 °C to 41 % at -50 °C. A significant fraction of columns exhibited hollowness on the basal facets, with 46 % and 40 % at initial gas temperatures of -40 °C and -50 °C, respectively.

We found that particles with $GMD$ larger than $10\,\mu m$ have a linear depolarisation ratio below 0.3. This is lower than atmospheric lidar measurements in mid-latitude cirrus but agrees well with lidar measurements in polar regions. Two temperature-dependent modes are found. For cloud chamber temperatures of -45 °C and lower, in most cases $\delta$ stays constant for increasing size. For cloud chamber temperatures between -45 °C and -39 °C, $\delta$ increases with increasing size up to about $20\,\mu m$. $\delta$ in both temperature ranges can be well reproduced with CGOM simulations of hollow columnar ice crystals using the tilted facet method and mean free paths for internal scattering to model ice crystal complexity. For ice crystals with $GMD$ below $10\,\mu m$, the strong increase in $\delta$ up to values of 0.45 with decreasing size is in good agreement with T-matrix simulations assuming spheroidal shapes. Recent IITM simulations for small and roughened hexagonal ice crystals were tested against our observations, but the comparison showed that the IITM simulations overestimate the measured $\delta$ by about 15 %. Adapting the model of ice crystal complexity further, for instance by including hollowness, may improve agreement with the measurement data.

We also investigated the link between the small-scale morphological complexity, measured using the optical complexity parameter $k_e$, and $\delta$, but no clear correlation was found. One reason can be that other morphological features, that are not constant during the experiments (like the particle shape), affect $\delta$. It is also possible that, even at the lowest supersaturation levels generated in the cloud chamber, ice crystals already have a baseline roughness, potentially limiting the range of $k_e$ so that a correlation cannot be observed with the current laboratory setup.

Understanding how size and morphological complexity influence the ice cloud backscattering linear depolarisation ratio is essential for interpreting atmospheric remote sensing data from instruments such as the Cloud-Aerosol Lidar with Orthogonal Polarization (CALIOP) of the CALIPSO mission, the Cloud-Aerosol Transportation System (CATS) lidar on the International Space Station (ISS) or the Atmospheric Lidar (ATLID) on the new Earth Cloud, Aerosol and Radiation Explorer (EarthCARE) satellite (Sassen and Benson, 2001; Pauly et al., 2019; Donovan et al., 2024). Additionally, our measurements of the optical

properties of small ice crystals can be useful for testing and validating optical particle models at the limit of the geometric optics approximation.

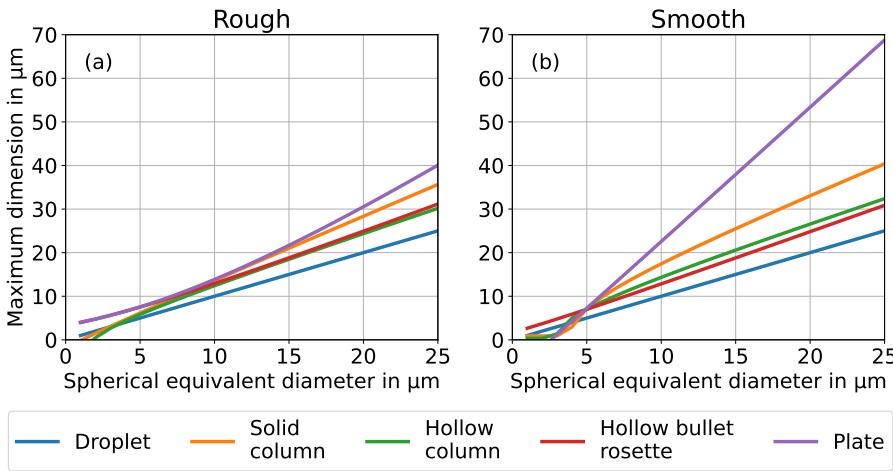

**Figure A1.** Ice crystal maximum dimensions of different habits are shown as a function of their spherical equivalent diameter for rough ice particles **(a)** and smooth ice particles **(b)**. The spherical equivalent diameter is estimated by integrating the light that is scattered by an ice crystal in the direction of the solid angle of the PPD-2K trigger field of view, based on a light scattering database by Yang et al. (2013) and Mie theory for droplets (Prahl, 2024). The maximum particle size can be up to 2.5 times the spherical equivalent diameter for rough plates.

## Appendix A:  Relation between spherical equivalent diameter and particle maximum dimension

Due to different light scattering properties in the direction of the trigger field of view of the PPD-2K, the relationship between the ice particle spherical equivalent mean diameter and the maximum dimension is influenced by the ice particle shape and complexity. The fraction of light scattered by water droplets of different sizes over the trigger optics solid angle is calculated with Mie theory (Prahl, 2024). It is compared to the intensity of light scattered in the direction of the trigger field of view by ice crystals based on light scattering properties from a database by Yang et al. (2013). We estimate a spherical equivalent diameter of 25 µm to be equivalent to a maximum dimension between 30 µm for a rough hollow column and up to 69 µm for a smooth plate. The maximum dimensions of ice crystals of different habits and water droplets are shown in Fig. A1a for rough particles and in Fig. A1b for smooth particles. The conversion factor ranges between 1.0 and 2.5. It can be noted that the spread in size conversion between the different habits is larger for smooth particles than for rough ones. Increasing particle roughness smooths out the characteristic scattering features of the particle habits, such as the 22° halo. Thus, the habit of pristine particles has a stronger effect on the size conversion than that of complex particles.

## Appendix B:  Additional numerical simulations

This section provides additional numerical simulations of the linear depolarisation ratio. T-matrix simulations of spheroids and CGOM simulations of solid and hollow hexagonal ice crystals are presented for multiple aspect ratios and for the wavelength of 552 nm that was used in the SIMONE-Junior instrument.

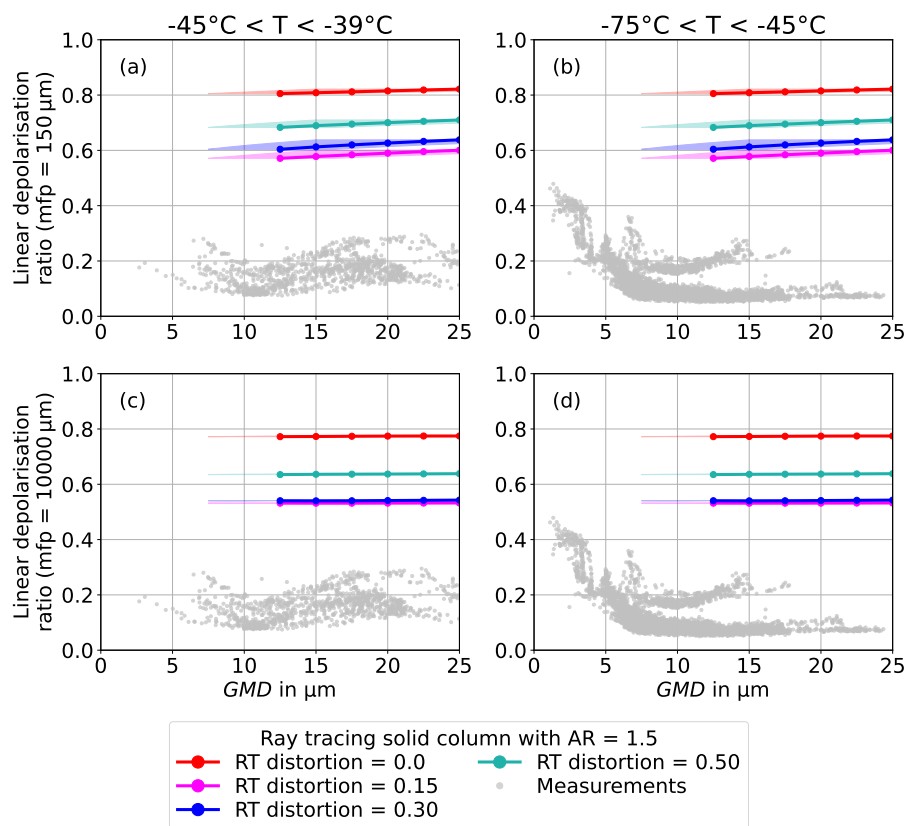

**Figure B1.** Linear depolarisation ratio from CGOM simulations at 178° near-backscattering direction as a function of geometric mean diameter ($GMD$) for solid columns (Macke, 2020). The aspect ratio (AR) is fixed at 1.5 and the distortion is varied between 0 and 0.5. The simulations in **(a)** and **(b)** use a mean free path of 150 μm **(c)** and of $10^4$ μm for high and negligible internal scattering, respectively **(d)**. For comparison, **(a)** and **(c)** include all measurements with initial gas temperatures between -45 °C and -39 °C and **(b)** and **(d)** between -75 °C and -45 °C as gray dots. The shaded areas mark the uncertainty in size conversion between the measurement data and the simulations. $\delta$ simulated for solid columns overestimates the measurement data.

## B1   CGOM simulations of solid columns

Fig. B1 shows the measurement data with CGOM simulations of solid columnar ice crystals with a fixed aspect ratio of 1.5. The simulations of solid columns always overestimate our observations, independent of varying surface roughness or mean free path. An increase in the distortion parameter up to about 0.3 leads to a decrease in $\delta$ for all investigated sizes. Beyond this point, $\delta$ increases again with increasing distortion for all investigated sizes.

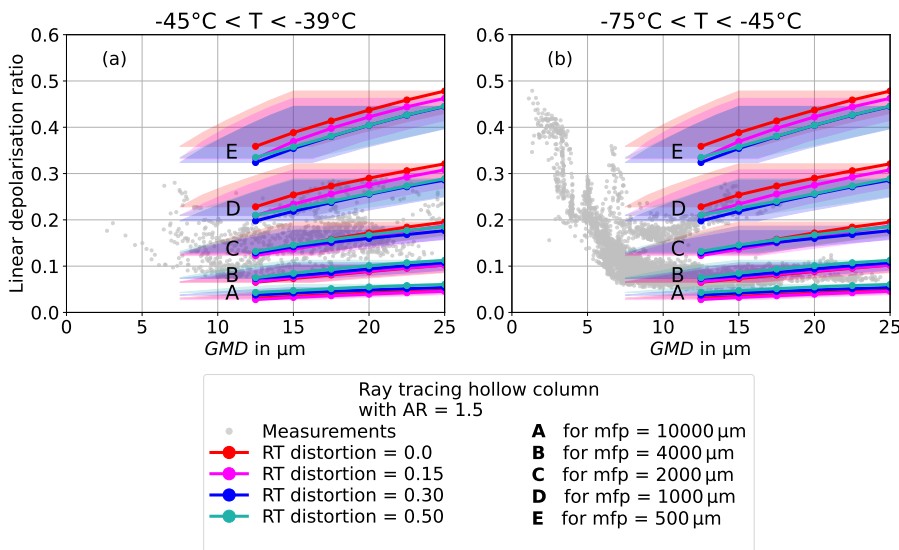

**Figure B2.** Linear depolarisation ratio as a function of geometric mean diameter ($GMD$). The measurement data (dots) are compared CGOM simulations of columns with hollow basal facets (entering 33.3 % of the column height from each basal facet) using mean free paths between $10^4$ µm (label **A**) and 500 µm (label **E**) (Macke, 2020). The experimental data at initial gas temperatures between -45 °C and -39 °C is shown in **(a)** and between -75 °C and -45 °C in **(b)**. The shaded areas mark the uncertainty in size conversion between the measurement data and the simulations.

## B2 Effect of mean free path on CGOM simulations of hollow hexagonal particles

Fig. B2a and Fig. B2b show CGOM simulations of hexagonal ice particles with hollow basal facets using mean free paths between $10^4$ µm (label **A**), indicating negligible internal scattering, and 500 µm (label **E**), indicating high internal scattering. A mean free path around 2000 µm (**C**) reproduces $\delta$ of the higher cirrus temperature range between -45 °C and -39 °C. Using a mean free path of about 4000 µm (**B**), the CGOM simulations reproduce $\delta$ of the lower cirrus temperature range between -75 °C and -45 °C. A decrease in mean free path increases the linear depolarisation ratio.

## B3 Effect of aspect ratio in CGOM simulations


Fig. B3a and Fig. B3b show that changes of the aspect ratio between 1.5 and 2.0 only have a minor effect on the simulated $\delta$ for the CGOM simulations of solid and hollow columns. This is the range of aspect ratios for columns seen on the microscope images of the formvar replica sampling at the AIDA cloud chamber. A distortion parameter of 0.3 and different mean free paths are used. The difference in $\delta$ is largest with 2.5 % between an aspect ratio of 1.5 and 2.0 for hollow columns. In addition,
T-matrix simulations of spheroids with aspect ratios between 1.5 and 2.0 are shown. Here, the aspect ratio has a larger effect on $\delta$ with differences of up to 20 % between aspect ratios of 1.5 and 2.0. Nonetheless, all three aspect ratios reproduce the size-dependence of the measurement data well in the size range of about 2 µm to 9 µm.

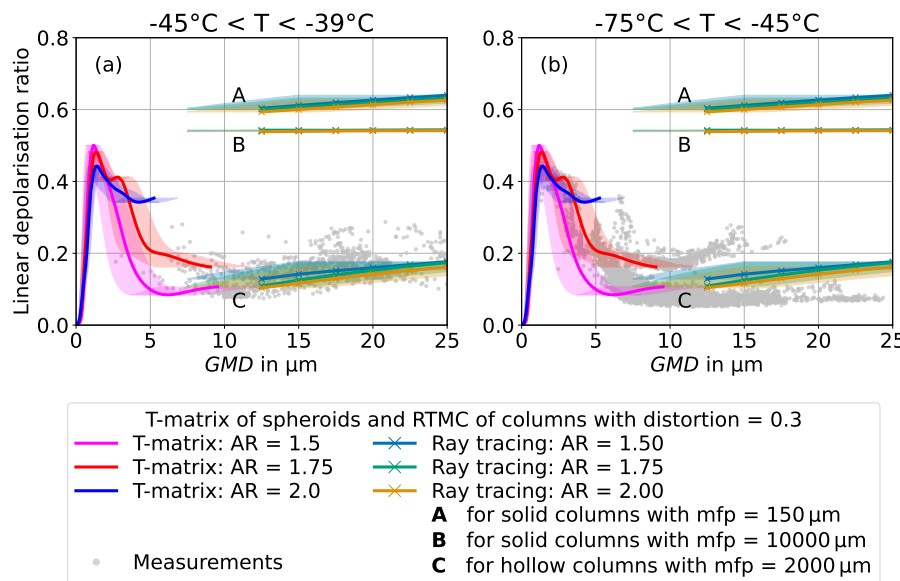

**Figure B3.** Linear depolarisation ratio as a function of geometric mean diameter ($GMD$). The measurement data (dots) are compared to T-matrix simulations of spheroidal particles at 178° near-backscattering direction (lines) (Leinonen, 2014) and to CGOM simulations of columns with a distortion of 0.3. The aspect ratio (AR) is varied between 1.5 and 2.0. Mean free paths of 150 μm (high internal scattering) and $10^4$ μm (negligible internal scattering) are used for solid columns (see **A** and **B**), and a mean free path of 2000 μm is used for solid columns (see **C**). For comparison **(a)** includes all measurements with initial gas temperatures between -45 °C and -39 °C and **(b)** between -75 °C and -45 °C as gray dots. The shaded areas mark the uncertainty in size conversion between the measurement data and the simulations. Changes in aspect ratio in the observed range between 1.5 and 2.0 only cause minor changes in $\delta$.

## B4 Effect of SIMONE-Junior wavelength of 552 nm

Fig. B4a and B4b show the simulated linear depolarisation ratio for solid and hollow columns at the different wavelengths

of 448 nm and 552 nm, which are used in the SIMONE and SIMONE-Junior instruments, respectively. The wavelength of 552 nm used in SIMONE-Junior during the RICE03 measurement campaign only causes minor changes in $\delta$, with a maximum deviation of 1.5 % in comparison to the wavelength of 488 nm that is used by the SIMONE instrument in all other measurement campaigns.

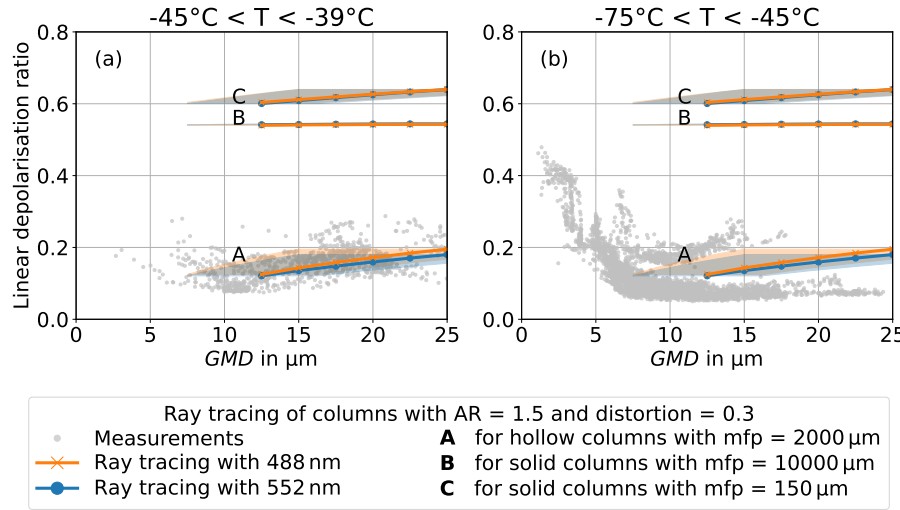

**Figure B4.** Linear depolarisation ratio from CGOM simulations at 178° near-backscattering direction as a function of geometric mean diameter ($GMD$) for solid and hollow columns (Macke, 2020) and for wavelengths of 488 nm and 552 nm, as they are used by the SIMONE and SIMONE-Junior instruments, respectively. The aspect ratio (AR) is fixed at 1.5 and the distortion is fixed at 0.3. Mean free paths of 150 μm (high internal scattering) and $10^4$ μm (negligible internal scattering) are used for solid columns (see **C** and **B**) and 2000 μm for hollow columns (see **A**). For comparison **(a)** includes all measurements with initial gas temperatures between -45 °C and -39 °C and **(b)** between -75 °C and -45 °C as gray dots. The shaded areas mark the uncertainty in size conversion between the measurement data and the simulations. Small changes in the used wavelength of the two instruments only cause minor changes in $\delta$ according to the CGOM simulations.

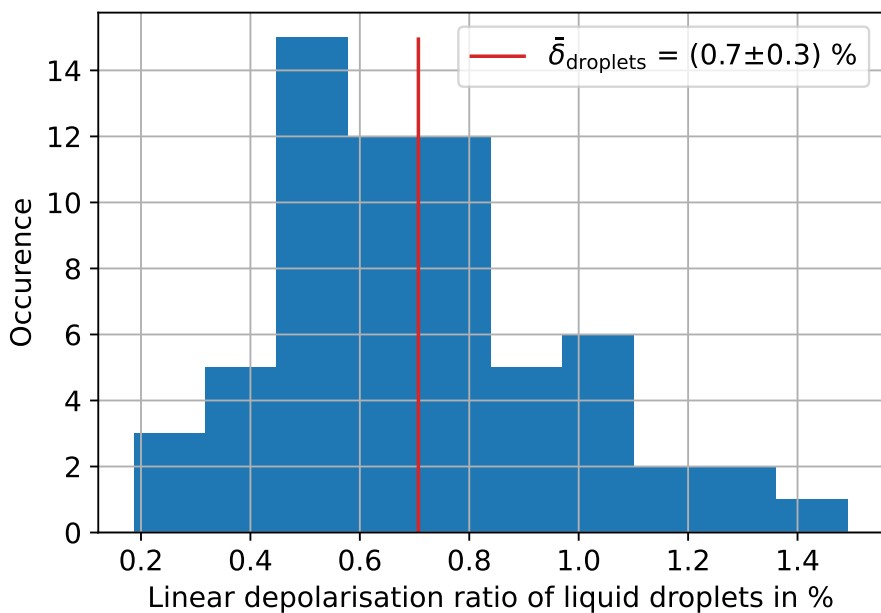

**Figure C1.** Histogram of the 10 s averaged linear depolarisation ratio measured during 12 supercooled liquid droplet cloud experiments in the AIDA cloud chamber during the RICE03 campaign at an initial gas temperature of -30 °C. The observed mean linear depolarisation ratio ($\bar{\delta}_{\mathrm{droplet}}$) is (0.7±0.3) %, which is well within the expected range taking the measurement uncertainty of 2.1 % from the calibration target measurements into consideration.

**Appendix C: Validation of linear depolarisation by observation of liquid droplets**

Fig. C1 shows a histogram of the 10 s averaged linear depolarisation ratio measured during 12 supercooled liquid droplet cloud experiments in the AIDA cloud chamber during the RICE03 campaign at an initial gas temperature of -30 °C. The linear depolarisation ratio of liquid and thus spherical droplets vanishes (Liou and Lahore, 1974). The observed mean linear depolarisation ratio of the liquid clouds ($\bar{\delta}_{\mathrm{droplet}}$) of (0.7±0.3) % thus supports the measurement uncertainty of 2.1 % derived from the calibration target measurements.

*Data availability.* All measurement data are available on RADAR4KIT: https://doi.org/10.35097/66tc7z2u0s2gf1fe

*Author contributions.* EJ and AH conceptualised the manuscript. MSc developed the SIMONE instrument. MSc planned and led the AIDA cloud chamber experiments. MSa provided the IITM simulations. RW operated and analysed the data of the FTIR. AH analysed the data and wrote the manuscript. All have read and commented on the paper.

*Competing interests.* None of the authors declare competing interests.

*Acknowledgements.* This work was funded by the Helmholtz Association's Initiative and Networking Fund, grant number VH-NG-1531. The authors are grateful to the AIDA staff for their support during the cloud chamber experiments.

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
