# Peer review of "Cloud Chamber Studies on the Linear Depolarisation Ratio of Small Cirrus Ice Crystals"

_EGUsphere, 2025_

## Author Comment (AC1)

**egusphere-2025-3515: Authors' Responses to Anonymous Referee #1**

*The authors thank anonymous referee #1 for the positive evaluation and comments. Responses to the different feedbacks are given below and modifications of the revised manuscript are detailed.*

The manuscript presents the linear depolarization ratio of small ice crystals ($< 70\,\mu m$) at 178° scattering angle under controlled laboratory conditions. The size and morphology of the ice crystals are well characterized in the laboratory. Therefore, the chamber studies present a valuable piece of information for atmospheric observations of cirrus clouds with lidar. Furthermore, it serves as constraint for optical models in the description of the complex shape of ice crystals. It is worth to analyze the AIDA measurements for this purpose and the topic is well suited for ACP. My concerns mostly regard the modelling approaches used to fit the observations. Therefore, it should be published after major revisions.

Major comments

1. A major point of critics is that you compare your measurement results to CGOM calculations for columns or even hollow columns, but only 19% (-40°C) and 40% (-50°C) of your observed ice crystals actually have a columnar shape. And of these, only 46% (-40°C) and 40% (-50°) are hollow columns (Tab. 2). It means that you found hollow columns only in 0.19*0.46 = 9% (-40°C) and 0.4*0.4= 16% (-50°C) of your ice crystal samples. Hollow columns represent only a minority of your lab measurements and it is not comprehensible why the CGOM modelling results of hollow columns should explain your data. The morphologically more complex particles which represent the majority of your ice crystals are more difficult to reproduce in optical models but are dominating your signals.

*Response: We agree with referee #1 that the majority of the observed ice particles are morphologically complex. In this work, we use columnar ice particles with different types of morphological complexity as a proxy for all ice particles for different reasons. First, for an ensemble of ice particles it is not feasible to model the full ice particle morphological complexity. Therefore, some degree of simplification must be made in order to keep the computational effort reasonable. We chose the hexagonal columnar particle shape because growth along the c-axis dominates at the investigated cirrus temperatures (Bailey and Hallett, 2009). Furthermore, the fractions of columnar and hollow columnar particles derived from the formvar replicas need to be considered as a lower bound. Limited image resolution as well as errors and artifacts from the formvar replication process restrict the identification of hexagonal particle shapes from the formvar replicas. This is detailed in an example microscope image of the formvar replicator stripe*

*of RICE01 experiment 20 at an initial gas temperature of −50 °C. The circles of different color highlight the ice particles which are classified as columnar particles (blue), columnar particles with hollow basal facets (red) and irregular particles (green). It cannot be excluded that more particles have columnar shapes than are clearly visible in the microscope image of the formvar replicas. The same is applies for the fraction of columnar particles with hollow basal facets. Line 286 is changed to "Therefore, the observed fraction of columnar ice particles and hollow columnar ice particles should be considered as a lower limit.".*

[Figure]

*Nonetheless, we conclude that improved representations of ice particle complexity in the optical models are needed. In this study, we would like to highlight that CGOM simulation of columnar particles with hollow prism faces can reproduce the linear depolarisation ratio of the cloud chamber grown ice particles.*

2. It seems that the IITM simulations were added later to the study. Sect. 2.4 firstly focuses only CGOM and T-Matrix and much later, you mention that IITM was used additionally. Also, Fig 8 and 9 are split between the models. Actually, the T-Matrix and the IITM both are used to describe the smaller particles up to 9 µm and for AR=1, IITM was extended up to around 17 µm which gives an overlap to CGOM. However, this overlap is not used in your discussions. Also, T-Matrix and IITM are not compared. I am wondering, if the increasing depolarization for small particles as seen for IITM is present in

the T-Matrix results as well. How does it look like if you start your calculations at GMD = 0 as done for IITM? A more comprehensive use of the 3 models and corresponding discussion is needed.

*Response: The T-matrix simulation range is increased to 82 size bins from 0.01 μm to 10 μm. The increase of δ for the smallest particle sizes is seen with the T-matrix method, too. Fig. 8, Fig. 10 and Fig. B4 are adapted accordingly (see figures below). Furthermore, the IITM simulations are introduced earlier in the methods section to give a better ordered overview of all three simulation methods. The discussion section of the light scattering simulations is revised to better compare the IITM simulations to the T-matrix simulations of spheroidal particles and to the CGOM simulation. Lines 367f. now read "While the simulated δ is lower than the CGOM simulations of solid hexagonal particles (see appendix B1), it overlaps with the CGOM simulations of hollow columns at low mean free paths (see appendix B2). Yet, the δ from the IITM simulations still generally overestimates the measurement data." and the following paragraph is added at line 370 "Similarities and differences in the evolution of δ with particle size from the IITM simulations are found in comparison to the T-matrix simulations. The sharp increase in δ for the smallest particle sizes in the IITM simulations is also seen in the T-matrix simulations. However, the peak of δ for the T-matrix simulations is at a lower size of about 2 μm in comparison to the IITM simulations. Furthermore, the subsequent plateau of δ is observed in both simulations, but at lower sizes and for lower δ in the T-matrix simulations in comparison to the IITM simulations. Further refining the ice crystal complexity model of the IITM simulations, for example, by incorporating hollowness, could enhance its agreement with the measurement data.".*

[Figure]

Figure 8

[Figure]

Figure 10

[Figure]

Figure B4

3. The T-Matrix method is used for spheroids. However, the shape of ice crystals is far away from a spheroidal shape. You need a strong motivation to use the spheroidal shape for the representation of the small ice crystals. The agreement of the results for the spheroidal shape might be pure coincidence. Because if it agrees already with spheroidal simulations (L447/448), then you don't need to ask for more complex optical models.

*Response: We agree that the spheroidal shape is a rough approximation of small ice particles. This is why the IITM simulations using more detailed rough-*

*ened hexagonal shapes are shown in comparison to the T-matrix simulations of ice particles with spheroidal shapes. We decided to show the simulations using the spheroidal shape, because the T-matrix implementation by Mishchenko and Travis (1998) is widely used in LIDAR applications (e.g. Nicolae et al., 2018; Veselovskii et al., 2010; Mishchenko et al., 2000). The limitation of the assumption of the spheroidal shapes in the simulations was additionally highlighted in line 199 "The assumption of spheroidal particle shapes is a rough approximation of the hexagonal shape of ice particles (Bailey and Hallett, 2009). Therefore, as a comparison to the T-matrix approach, which is widely used in LIDAR applications (e.g. Nicolae et al., 2018; Veselovskii et al., 2010; Mishchenko et al., 2000), light scattering simulations of hexagonal ice crystals with the Invariant-Imbedding T-matrix Method (IITM) for size parameters of up to 158 are additionally compared to the measurement data.".*

4. Do you have to consider possible orientation/alignment effects of the ice crystals? Maybe not, because you're observing them from the side in a turbulent environment. However, in lidar, specular reflection caused by horizontally oriented ice crystals lead to much lower depolarization ratios in vertically pointing lidars. To avoid the effect of specular reflections in cirrus clouds, most lidars are tilted to an off-zenith angle of 3-5°.

*Response: Inside the AIDA cloud chamber there is a mixing fan that was run during the experiments. The sensing volume of SIMONE is in the vicinity of the mixing fan (see black circle in figure below). Therefore we can assume turbulent environment with random particle orientation in the sensing volume of SIMONE. Line 65 is updated to "Well-mixed conditions and random particle orientations persist due to operating a mixing fan in the cloud chamber.".*

[Figure]

Minor comments:
An outline at the end of the introduction is common to guide the reader through the manuscript. Furthermore, one or two introductory sentences at the beginning of each section would be helpful. It would help to keep the track of your study.

*Response: Introductory sentences in the beginning of each section are updated to the manuscript. Furthermore, the following outline is added at the end of the introduction at line 59 "The microphysical and optical instrumentation used to measure the microphysical and optical properties of the cloud chamber grown ice particles are introduced in sections 2.1 and 2.2. The experimental procedures and the numerical simulations are detailed in sections 2.3 and 2.4. Section 3.1 contains a detailed analysis of the microphysical properties, such as size, small-scale complexity and hollowness, of the cloud chamber grown ice particles. Hereafter, the microphysical properties size (section 3.2) and small-scale complexity (section 3.3) are linked to the linear depolarisation ratio, with the main outcome being the evolution of the linear depolarisation ratio as a function of the particle size. Section 3.4 compares the experimental results to different numerical T-matrix and ray tracing light scattering simulations. In section 4, the cloud chamber measurements are compared to previous cloud chamber and atmospheric observations and the limitations of the different numerical simulations and of the ice particle morphology representation are discussed. A summary of the study is provided in section 5.".*

Eq 1+4: Why do you name it delta_parallel and not just delta?A sketch of the

setup and the involved instruments would be nice. I agree that it is already presented in previous publications, but a visualization (even very schematic) would help to better follow the methods section.

*Response: We adapted equations (1) and (4) to use the symbol $\delta$ to refer to the linear depolarisation for incident light with linear polarisation parallel to the scattering plane. This is now consistent throughout the manuscript. Furthermore, the manuscript is updated with a schematic setup of the SIMONE instrument (see Fig. 1a).*

Fig 1 Does the scattering pattern depend on particle orientation? And how do you deal with randomly oriented regular particles?

*Response: Due to turbulent conditions in the measurement chamber of PPD-2K (caused by the nozzle focusing the particle stream), the single particles detected with PPD-2K are randomly oriented. The particle orientation alters the scattering patterns, and at random orientations, for instance the main arc in the diffraction pattern becomes curved when the ice crystal is not parallel or perpendicular to the incident laser beam (see also Vochezer et al. (2016); Ulanowski et al. (2006)). Generally, hexagonal particles at random orientations still return the described Fourier maxima in the diffraction patterns (see examples below for Fast Fourier Transform (FFT) maxima of order 2, 3, 4 and 6). Nonetheless, at specific orientations, the Fourier analysis method may misclassify hexagonal particles as irregular particles. Out of 100 random ice particles, which were classified as irregulars by the Fourier method, 11 hexagonal ice particles with bent arcs could be manually identified. This is another reason why the fraction of columnar and plate-like particles should be seen as a lower limit. The following sentence was added to line 126 "Furthermore, the fraction of columns and plates should be seen as lower limit because the random particle orientation can lead to bent arcs in the scattering patterns of pristine hexagonal ice particles, which may occasionally lead to false interpretation as irregulars.".*

[Figure]

L119-149: These paragraphs should be carefully checked again, because there are some small careless mistakes, e.g., twice a reference to Fig. 1c, a different uncertainty of sigma in Fig. 1e and the text (L135), in bracket references not correctly formatted, and some rather slang formulations.

*Response: We thank the reviewer for spotting the mistakes. The section is checked again and the reference and uncertainty of $\sigma$ errors are corrected. Line 119-130 are rephrased and now read "Furthermore, the particle shape is derived from the diffraction patterns. If there are clear maxima in the azimuthally integrated polar profile (Mie fringes) the particle is considered spherical and is classified as a droplet (see Fig. 2d). Otherwise, a discrete fast Fourier transform (FFT) of the polar integrated azimuth intensity profile is performed for the shape identification following the procedure of Vochezer et al. (2016). Maximum Fourier coefficients of order 2 and 4 indicate a columnar shape (Fig. 2c) and maximum Fourier coefficients of order 3 and 6 indicate a hexagonal plate (Fig. 2a). If the maximum coefficient is of a non-symmetric order the particles are interpreted as irregulars (Fig. 2b). It needs to be noted that if a particle is classified as irregular it does not necessarily mean that it has an irregular shape. Irregular diffraction patterns may also result from sufficiently roughed columnar or hexagonal ice particles, which do not show their usual, distinct diffraction patterns. Furthermore, the fraction of columns and plates should be seen as a lower limit because the random particle orientations can lead to bent arcs in the scattering patterns of pristine hexagonal ice particles, which may occasionally be falsely interpreted as irregulars. Out of 100 random ice particles classified as irregulars by the Fourier method, 11 hexagonal ice particles with bent arcs were manually identified. In this work, the measured single particle scattering infor-*

*mation is converted into particle size distributions using 50 bins in a size range between about 7 μm and 70 μm depending on the campaign-specific size calibration. The particle size distributions are averaged over 10 s and a log-normal size distribution function is fitted to obtain the geometric mean diameter (GMD) and the standard deviation of the logarithm of GMD (σ).".*

In Sect. 2.3, I don't like the formulations like "minute 25" in a written text. A suggestion would be to write "at t=25 min". Furthermore, Fig 2 should be introduced once before presenting the procedure. Is it a typical experiment to give an example? Or were all experiments done like this?

*Response: We adapted the reviewers suggestion. Line 152 is updated to "The temporal evolution of a typical experiment is shown in Fig. 3.".*

L186: What is mu? Overall, this note is a bit tricky to understand.

*Response: This is a typo. The data is used if the GMD determined by the log-normal fit is at most 20 % smaller than the center of smallest bin of the PPD-2K size range to ensure an accurate retrieval of GMD. Line 186 is corrected to "The GMD determined by a log-normal fit is at most 20 % smaller than the center of smallest bin of the PPD-2K size range to ensure an accurate retrieval of GMD: $1.2 \cdot d_{\min} \leq GMD$ .".*

L215: What exactly do you mean with spherical equivalent diameter? Which quantity is equivalent to a sphere (volume, surface, . . . )?

*Response: The spherical equivalent diameter d is the diameter of a sphere that scatters the same intensity in the direction of the PPD-2K trigger field of view as the recorded particle. The following sentence was added to the manuscript for clarification: "PPD-2K measures the trigger intensity I of each particle. To relate I to the particle size, the spherical equivalent diameter d is used, which is the diameter of a sphere that scatters the same intensity in the direction of the PPD-2K trigger field of view at polar angles between 7.4° and 25.6° as the recorded particle.".*

And why you are using the maximum dimension of a spheroid (L215) or the column length (L228) for comparison to the spherical equivalent diameter?

*Response: The conversion between the spherical equivalent diameter measured with PPD-2K and the particle sizes in the different particle models of the optical simulations is non-trivial and therefore an approximation must be used. In this work, we use the maximum dimension of a spheroidal and the column length of a columnar particle as characteristic lengths, which is a commonly used approach when analysing the optical properties of complex shaped particles (e.g. Liu et al., 2014). Following the comments of Darrel Baumgardner and Zbigniew Ulanowski an uncertainty estimation for the size conversion was performed and added as*

*a shaded area to the simulations. According to the calculations in Fig. A1, the particle maximum dimension of rough columnar or bullet rosette shapes ice particles can be up to 40 % larger than the spherical equivalent diameter, depending on the particle type, because the transmission component of the scattered light in the forward direction is almost always smaller for nonspherical particles than for spheres with the same size. Furthermore, in case of a columnar particle, the maximum dimension can be larger than the column length, depending on the aspect ratio (Um et al., 2015). For example, the maximum dimension of a column with an aspect ratio of 1.5 is about 30 % larger than its length. This uncertainty is also added as shaded area to figures 9, 10, B1, B2, B3 and B4. Lines 236f. are updated to "The conversion between the particle sizes from the numerical simulations and the spherical equivalent diameter measured with PPD-2K is non-trivial and therefore an approximation is needed. In this work, we use the maximum dimension of the spheroidal particles and the column length of the hexagonal particles, which is common approach when analysing the optical properties of ice particles (Liu et al., 2014). To estimate possible errors, the spherical equivalent diameter measured with PPD-2K is calculated for different complex shaped ice particles in Fig. A1. The maximum particle dimension can be up to 40 % larger than the spherical equivalent diameter for rough columnar and bullet rosette shaped ice particles, because the transmission component in the forward direction of the scattered light is almost always smaller for nonspherical particles than for spheres with the same size. However, the column length of columnar particles can also be smaller than the maximum particle dimension, depending on its aspect ratio (Um et al., 2015). For example, a columnar particle with an aspect ratio of 1.5 has a maximum dimension, which is about 30 % larger than its length. These uncertainties in size conversion are shaded in the figures where the results of the numerical simulations are compared to the measurements with PPD-2K.". In addition, the shaded areas is labelled as uncertainty in size conversion in the corresponding figure captions.*

L234: Please provide the refractive index.

*Response: The refractive index (n,k) that was used for the IITM simulations is $n = 1.3116$ and $k = 1.49 \cdot 10^{-9}$. It is added to the manuscript.*

It would be recommended to better link Fig 4 with Tab 2 as some images belong to cases presented in the table.

*Response: A better link is between figure 4 and table 2 is added. Line 267f. is updated to " Example microscope images of formvar replicas from experiment 29, 32, 38 and 40 of the RICE04 campaign are shown in Fig. 4e,d,c,a. Due to the small sample number, the fraction of columnar particles and hollow columnar particles seen in Fig. 4 is not representative. Nonetheless, the general occurrence of hollowness on the prism faces at both cirrus temperatures and the smaller particles sizes at -50 °C in comparison to -40 °C can be observed on the microscope images.".*

The caption of Fig 4 is quite long. You may consider to add some information, e.g., the supersaturation, already to the images.

*Response: The caption was shortened and the temperature and humidity information was added to the images.*

L275 and Fig 5: You split your results by campaign. It is important for your analysis. However, for the reader it is a bit confusing as I don't see the differences between RICE 01 – 03. An overall number for each regime might be sufficient. It is up to you how to best harmonize the description.

*Response: The figure and paragraph have been adapted and now show and analyse the correlation coefficients between the optical small-scale complexity and the linear depolarisation ratio combined for the three measurement campaigns. Lines 316f. now read "For the smaller three particle size ranges of $(5 \pm 1)\,\mu m$, $(10 \pm 1)\,\mu m$ and $(15 \pm 1)\,\mu m$ $|R| \leq 0.13$ indicates no or weak correlation between $k_e$ and $\delta$. Only for the largest size range of $(20 \pm 1)\,\mu m$ a moderate correlation of $R = 0.54$ is observed. The scarcity of particles in this largest size group during the RICE02 measurement campaign can be attributed to the lower initial gas temperature of $-50\,°C$, compared to RICE01 and RICE03, which included experiments at $-50\,°C$ and $-40\,°C$. Higher growth temperatures promote the formation of larger ice particles through increased growth rates at the same relative humidity (Bailey and Hallett, 2009).".*

L353/354: Please add a reference to the appendix as well.

*Response: The reference is added and the sentence is rephrased and moved at the end of the section for clarity. Lines 358-359 now read "CGOM simulations of solid columns overestimate $\delta$, as demonstrated in appendix B1, where ice crystal complexity is varied with distortion parameters from 0 (pristine) to 0.5 (highly complex).".*

L392: Actually, it is not in line with your chamber studies, because in the colder temperature regime (CIRRUS-ML, -58 – -63°C), you observe a constant depolarization ratio and then a increase towards smaller particle sizes. The difference in the scattering angle and the possible implications should also be mentioned in the conclusion.

*Response: Indeed, the sentence was adapted and the different scattering angles were mentioned. Lines 392-393 now read "The $\delta$ observed during CIRRUS-ML measured at the direct backscattering direction largely exceeded the $\delta$ observed at the AIDA cloud chamber at the near backscattering direction of 178°.".*

L460: A better reference for EarthCARE would be Donovan et al., AMT 2024 as they describe the lidar measurements and the retrieval of ice crystal properties. References to Sato and Okamoto are missing. In several publications, they discussed the modelling of the depolarization ratio of cirrus clouds. Furthermore, you can search for more lidar observations of the depolarization ratio of cirrus clouds.

*Response: References to Donovan et al. (2024) were added for EarthCARE in line 460. Furthermore, in the introduction the optical particle model by Sato et al. (2019) was included in the overview. Lines 29f. now read "In addition, global cirrus cloud particle habit fractions were derived from CALIOP satellite LIDAR data using the physical particle model, an improved geometric optics ray tracing methods which includes multiple scattering effects (Sato and Okamoto, 2023).". Furthermore, additional LIDAR studies have been added to the overview in the discussion section. Lines 378f. have been updated to "Most other lidar studies of mid-latitude cirrus reported δ in the range between approximately 0.2 and 0.5 (e.g. Wang et al., 2008; Kim et al., 2014; Urbanek et al., 2018; Manoj Kumar and Venkatramanan, 2020; Li and Groß, 2021; Gil-Díaz et al., 2024; Li and Groß, 2025). ".*

The plots of Fig B2 and B4 are swapped. It really created a lot of confusion for me until I've got that the figures are not the ones explained in the caption.

*Response: Corrected. We are sorry for the confusion.*

Technical Corrections

L 436: Skip the word "about". Or is the number of experiments not sure?

*Response: The experiment number is 47. We removed the word "about" in line 436.*

Fig B3: According to the figure, C is shown for hollow columns not solid ones. The caption is probably wrong.

*Response: The figure caption of Fig. B3 is correct. Plot C shows δ of RTMC simulations of hollow columns of aspect ratios 1.50, 1.75 and 2.00 .*

**Authors' Responses to Darrel Baumgardner**

This is a very nicely written discussion of what I consider to be a very important subject matter, i.e. interpretation of polarization ratios derived from both passive and active remote sensors. The authors have conducted very thorough research and have approached the problem from multiple directions in order to achieve some semblance of closure. What I think is missing, or perhaps has been overlooked, is a thorough uncertainty analysis, not only of the measurements but of the models, as well.

*Response: The authors thank Darrel Baumgardner for the positive evaluation and feedback. Detailed responses to the comments are provided below. We agree that an improved uncertainty analysis is needed for this study and have added a discussion of the depolarisation measurement uncertainty (see reply to comment below). In addition, the uncertainty in size conversion of the simulation results has been estimated and added as shaded area to the figures of the simulation results. Please see the response to the comment by anonymous referee #1 for more information about the conversion between the spherical equivalent diameter and the characteristic length of the simulated ice particle and the corresponding uncertainty estimation.*

There are also some open questions that I have about the SIMONE measurement system that are related to my request for the analysis of uncertainties. Before evaluating the differences in the models and measurements, I think that it is imperative to discuss uncertainties. I am skeptical of the depolarization uncertainty that is stated as 1.4%, based solely on what is stated as calibration cycles. I didn't read the Schnaiter (2012) paper but my opinion is that referring the reader to this paper is insufficient due to the importance of this derived ratio to the main focus of the paper.

*Response: We agree that the calibration of the SIMONE instrument is of great importance to the manuscript. We now added the calibration procedure of measurement campaign RICE03 to give details on how the measurement uncertainty of the SIMONE instrument is derived (see figure below). The corresponding measurement uncertainty is 2.1 %, based on the standard deviation of the linear depolarisation ratio during the calibration procedure.*

[Figure]

Time in UTC on 15 December 2014 in HH:MM

*The observation of spherical particles enables an additional validation of the linear depolarisation ratio measured with the SIMONE instrument because the linear depolarisation approaches zero for spherical particles without multiple scattering effects (Liou and Lahore, 1974). During the RICE03 campaign, supercooled liquid droplets were observed at initial gas temperatures of -30 °C in the AIDA cloud chamber. A histogram of the linear depolarisation ratio of 12 clouds of supercooled liquid droplets is shown in the figure below. The SIMONE instrument measured a linear depolarisation ratio of (0.7±0.3) %. This is well within the expected measurement uncertainty of 2.1 % derived from the calibration with the scattering target.*

[Figure]

*Lines 71-83 now read "During the calibration process, which is identical for SIMONE and SIMONE-Junior, the polarisation of the incident light is switched between parallel and perpendicular orientation with reference to the scattering plane. A calibration factor is multiplied to the channel, which records the intensity with parallel polarisation, in order to obtain the same linear depolari-*

*sation ratio for the linearly polarised incident light at both orientations. Fig. 1b shows the calibration process exemplary for measurement campaign RICE03, where a calibration factor of 1.4908 was derived for SIMONE-Junior. The calibration factor takes into account the different gains of the detectors, different losses of the polarisation filtering and effects of possible differences in alignment of the detectors. The measurement uncertainty of δ is 2.1 %, derived as the standard deviation of the linear depolarisation ratio of the scattering target measured during the calibration process. Experiments with supercooled liquid droplets were conducted at the AIDA cloud chamber to validate the measurement uncertainty because the linear depolarisation ratio of liquid and thus spherical droplets vanishes (Liou and Lahore, 1974). The mean linear depolarisation ratio of 12 experiments of supercooled liquid clouds at an initial gas temperature of -30 °C is (0.7±0.3) %. The derivation from the theoretical value of 0 % is well below the measurement uncertainty of 2.1 % derived from the calibration with the scattering target. Additional information about the supercooled liquid droplet experiments is provided in appendix C". The validation of the measurement uncertainty with supercooled liquid clouds is added to the manuscript as appendix C1.*

Although it is not clearly explained in this paper, I believe that the SIMONE does not measure single particle scattering but detects scattering from an ensemble of particles. That being said, what impact do the following have on the derived polarization ratios: 1) number concentration, 2) size distribution, 3) crystal orientation, 4) spatial distributions within the sample volume (if not random, then how will this impact the measurement?).

*Response: It is correct that SIMONE detects the scattering of multiple particles. This is now highlighted in line 62, which is updated to "The linear depolarisation ratio (δ) of an ensemble of ice particles in the cloud chamber is measured with the SIMONE (Streulichtintensitätsmessungen zum optischen Nachweis von Eispartikeln - Scattering Intensity Measurements for the Optical Detection of Ice Particles) instrument (Schnaiter et al., 2012)." 1) For atmospheric LIDARs, multiple scattering can affect the linear depolarisation ratio (Shcherbakov et al., 2023). On the 2 m path length of SIMONE during the AIDA experiments, a maximum optical density of 0.47 is reached at a particle concentration of $20.4\,\text{cm}^{-3}$ and a geometric mean diameter of $19.0\,\mu\text{m}$. This leads to a minimum transmission of 62 % and thus a maximum possible contribution of multiple scattering of 38 % during the cloud chamber experiments. However, the transmission of the scattered light is above 90 % for 89.9 % of the measurement data. Thus, below 10 % of the intensity detected with SIMONE can be affected by multiple scattering effects for 89.9 % of the measurement data (see figure of histogram of optical depth below). Line 81 now reads "Based on an estimation of the optical depth from the particle microphysics measurements, below 10 % of the intensity detected with SIMONE can be affected by multiple scattering effects for 89.9 % of the measurement data.".*

[Figure]

*2) The particle size distribution is measured with PPD-2K simultaneously to the measurements of the linear depolarisation ratio and treated accordingly in the data analysis. Due to the growth conditions in the AIDA cloud chamber during expansion experiments, the particle size distribution can be approximated by a log-normal particle size distribution. This particle size distribution is also used for the numerical simulations that are compared to the measurement data. 3) The particle orientation within the sample volume is assumed to be random due to the proximity of the sample volume to the cloud chamber mixing fan. 4) The spatial distribution in the sample volume can be assumed to be homogeneous due to well-mixed conditions in the AIDA cloud chamber, which are assured by operating a mixing fan during the experiments. The lower concentration limit of $0.3\,cm^{-3}$, which is applied to the data analysis, ensures a minimum average number of 20 particles in the measurement volume of SIMONE. In line 65, "Well-mixed conditions and random particle orientations persist due to operating a mixing fan in the cloud chamber." is added.*

My second concern is that there does not appear to be any mention of the difference between 178 degrees and 180 degrees backscattering on the polarization ratios. From my own modeling of polarized backscattering from single particles, there can be quite a difference in polarization ratios when looking at 178 degrees versus 180 degrees. Was a sensitivity analysis done in order to evaluate the differences between lidar at 180 and SIMONE and models at 178 for ensembles? Once a detailed uncertainty analysis is include in this submission, I will be ready to accept it for publication.

*Response: A sensitivity analysis was performed and is shown in Figure 10 of the manuscript. The estimated difference between the linear depolarisation ratio at 178° and 180° depends on the used numerical simulations and the particle size. While the estimated difference for the different t-matrix and CGOM simulations is below 0.04, it can be up to 0.17 for the IITM simulations, where it increases with particle size. Therefore, as mentioned in the discussion section, it cannot be entirely ruled out that the slightly lower δ values observed in AIDA, compared*

*to atmospheric lidar, are due to the deviation from the direct backscattering direction.*

**Authors' Responses to Zbigniew Ulanowski**

This is a potentially very valuable contribution to cloud property retrieval using polarisation lidar. It is supported by extensive measurements using a whole array of instruments to not only measure depolarisation but also to characterise chamber-generated ice particle clouds. However, I concur with Darrel Baumgardner's concern (https://doi.org/10.5194/egusphere- 2025-3515-RC2) about the accuracy of the measurements and possible bias due to the difference between backscattering at the exact 180 and 178 degree angles. For example, differences may arise due to the phenomenon of coherent backscattering, which tends to create a sharp peak at 180 degrees. This also raises the possibility that the methods used are unable to accurately represent depolarisation at or near to the backscattering direction. In particular, it has been shown that the geometric optics method used here produces erroneous results in this context.

*Response: The authors thank Zbigniew Ulanowski for his assessment and feedback. We would like to highlight that our cloud chamber observations were conducted at 178° backscattering and are compared with numerical simulations at 178° backscattering. Thus, no bias from the detection angle is expected in the comparison to the numerical simulations. However, we agree with the concern about our comparison to atmospheric lidars at exact backscattering and refer to the answer to the comment of Darrel Baumgardner about the possible bias due to the difference between backscattering at the exact 180° and 178° direction. We acknowledge that the reliability of the CGOM method used for calculating polarisation properties has been questioned as it is detailed in the discussion section of the manuscript. However, the T-matrix and IITM methods take coherent backscattering into account, which can be seen for example in the size dependence in Fig. 11a,d. Furthermore, we would like to highlight our conclusion that an optimised, physically realistic ice particle model is needed.*

I also have serious misgivings about the methods used to size the ice particles. The authors write that they use a simple expression for size: $D = aI^b$, where I is the scattered flux, and $a$ and $b \approx 0.5$ are constants, citing Vochezer et al. (2016) who in turn cite the paper by Cotton et al. (2010) containing this expression. Then the authors state that the unknown "factor a is calibrated with spherical droplets". Either some details have been omitted from the description of this method, or it is inaccurate. The original paper by Cotton et al. (2010) goes on to point out that "the scattered flux from the sphere is larger by a factor of 3.0 [compared to ice particles]". Hence the omission of this correction would lead to undersizing of ice crystals by a factor of $\sim 1.7$. While the magnitude of this correction will vary with the detection geometry (range of scattering angles that are detected), in general it cannot be neglected. Has an attempt been made to extablish the magnitude of this correction factor for the specific geometry of the PPD-2K instrument that is reported to have been used here? If not, the sizing data may be invalid.

*Response: We agree that the expression for the droplet size should be cited from the original reference by Cotton et al. (2010) and added this reference to the manuscript. The factor of a is calibrated for each campaign and determined from a function fit of measured droplets. This gives accurate sizing of spherical droplets. The size measure of the ice particles is the spherical equivalent diameter, which is the diameter of a sphere that scatters the same light intensity in the direction of the PPD-2K trigger field of view as the recorded particle. This additional information was missing and has been added to the methods section. Lines 99-109 are adapted to "PPD-2K measures the trigger intensity I of each particle. To relate I to the particle size, the spherical equivalent diameter d is used, which is the diameter of a sphere that scatters the same intensity in the direction of the PPD-2K trigger field of view at polar angles between 7.4° and 25.6° as the recorded particle. d is calculated from the trigger intensity I using the following equation (Cotton et al., 2010):*

$$d = a \cdot I^b \qquad (1)$$

*where calibration coefficient a depends on the laser power and PMT gain and calibration coefficient $b = 0.522$ depends on the trigger geometry and must be around 0.5 because the scattered intensity is proportional to the geometric cross section of the particle (Vochezer et al., 2016). For each measurement campaign, the factor a is calibrated with spherical droplets where the size is determined comparing the Mie fringes of the diffraction patterns to Mie theory (see Vochezer et al. (2016) for details). The ice crystal maximum dimension is estimated to be approximately 1.0 to 2.5 times larger than the spherical equivalent diameter depending on the crystal shape and complexity (see appendix A).".*

*Indeed, an attempt has been made to relate the spherical equivalent diameter of ice particles to their maximum dimension using a light scattering database by Yang et al. (2013) because the transmission component in the forward direction of the scattered light is almost always smaller for nonspherical particles than for spheres with the same size. The relation is shown in the appendix of the manuscript in figure A1 and shows that the particle maximum dimension can be up to 2.5 times the spherical equivalent diameter for randomly oriented pristine ice particles and up to 1.6 times the spherical equivalent diameter for randomly oriented rough ice particles. The conversion uncertainty has been added to the figures as shaded areas. Please see the reply to the question of Darrel Baumgardner for further information. In addition, we would like to highlight that optical particle counters generally have this limitation when detecting non-spherical particles.*

Finally, the last statement in the abstract gives the impression that the authors somehow aim to eat their cake and eat it too: can the study simulateously contribute to "interpretation of thelinear depolarisation ratio of small ice crystals in active remote sensing and evaluating the performance of state-of-the-art optical particle models"? I would claim that it should be either, or: either

the measurements are used to verify the models, or to provide interpretation of depolarisation as a remote sensing tool. The only exception to this dichotomy would be if evidence of complete closure between the ice properties, depolarisation, and model results was provided, but this is not seriously attempted. So the aims should be clarified, and appropriate emphasis adopted.

*Response: The last sentence of the abstract has been adapted and now reads "These results are important for the interpretation of the linear depolarisation ratio of small ice crystals in active remote sensing or can be used for evaluating the performance of state-of-the-art optical particle models, especially for small size parameters below 100.".*

[revised manuscript text omitted]